# Fast-charging high-energy lithium-ion batteries via implantation of amorphous silicon nanolayer in edge-plane activated graphite anodes

Namhyung Kim [1], Sujong Chae[1], Jiyoung Ma[1], Minseong Ko[1,2] & Jaephil Cho[1]

As fast-charging lithium-ion batteries turn into increasingly important components in forth-coming applications, various strategies have been devoted to the development of high-rate anodes. However, despite vigorous efforts, the low initial Coulombic efficiency and poor volumetric energy density with insufficient electrode conditions remain critical challenges that have to be addressed. Herein, we demonstrate a hybrid anode via incorporation of a uniformly implanted amorphous silicon nanolayer and edge-site-activated graphite. This architecture succeeds in improving lithium ion transport and minimizing initial capacity losses even with increase in energy density. As a result, the hybrid anode exhibits an exceptional initial Coulombic efficiency (93.8%) and predominant fast-charging behavior with industrial electrode conditions. As a result, a full-cell demonstrates a higher energy density ($\geq$1060 Wh l$^{-1}$) without any trace of lithium plating at a harsh charging current density (10.2 mA cm$^{-2}$) and 1.5 times faster charging than that of conventional graphite.

[1] School of Energy and Chemical Engineering Green Energy Materials Development Center, Ulsan National Institute of Science and Technology (UNIST), Ulsan 44919, Republic of Korea. [2] Department of Metallurgical Engineering, Pukyong National University, Busan 48547, Republic of Korea. Correspondence and requests for materials should be addressed to M.K. (email: msko876@pknu.ac.kr) or to J.C. (email: jpcho@unist.ac.kr)

With rapidly increasing demands to reduce the charging time of portable electronics and electric vehicles, achieving fast charging in lithium-ion batteries (LIBs) with high-energy density has been intensively pursued as the most important criterion for practical utilization in forthcoming applications. However, conventional graphite anodes have been limited in terms of fast charging by the metallic lithium plating phenomenon caused by its sluggish intercalation kinetics[1–3] and low lithiation voltage (0.08 V vs. Li/Li$^+$)[4]. In particular, at a high charging current, large anode polarization pushes the graphite potential to the threshold for metallic Li deposition[5]. The deposited Li also undergoes electrical isolation[6] and reaction with the electrolyte[7, 8], which increase the internal resistance and decrease the energy density[9, 10]. In general, it is known that conventional graphite electrodes suffer from Li deposition giving rise to rapid capacity fading, even at a charging current density of 4 mA cm$^{-2}$[11, 12].

Alternatively, silicon has been highlighted as a feasible candidate with a gravimetric capacity (3572 mAh g$^{-1}$) that is 10 times that of graphite and a safer lithium-alloying potential (0.22 V vs. Li/Li$^+$), which prevents undesirable Li plating[4, 13]. Nevertheless, its fast charge performance deteriorates owing to the extreme volume change (> 300%) and poor electrical conductivity (~10$^{-4}$ S m$^{-1}$), resulting in disintegration of the electrode and a high charge-transfer resistance[13–17].

For these reasons, various strategies have been explored to enhance the rate property of both the anodes via the designing of porous structures (e.g., KOH etched graphite[12], MoO$_x$-catalyzed porous graphite[18], or 3D mesoporous silicon[19, 20]) and composites with a conductive matrix (e.g., graphite with vapor-grown carbon fibers[21], carbon-nanotube–graphene anchored silicon[22, 23], or silicon–metal composites[24, 25]). However, even though such approaches can enhance the reactivity with Li$^+$ ions and electron transport through largely exposed surface areas and conducting material, their application in high-energy LIBs are still hindered by the poor initial CE, low tapping density, and excessive proportion of the conducting matrix.

In order to realize high-energy density in electrodes, the fabrication process should be designed by taking into consideration the industrial conditions involving high areal capacity loading (≥3.0 mAh cm$^{-2}$) and high electrode density (~1.6 g cm$^{-3}$) under the limited amount of binder (≤3 wt%) and conductive materials (≤1 wt%). Since such electrode parameters are regarded as major limiting factors of the capability for fast charging[26], electrochemical evaluation under these conditions is necessary to evaluate the feasibility of achieving fast-charging characteristics while maintaining high-energy density. However, most previously reported anodes designed for fast (dis)charging were simply studied under mild electrode conditions to demonstrate plausible electrochemical performance by using low areal mass loading, uncalendered electrodes, and excessive amounts of binders and conductive agents, which, in turn, bring about low-energy density. Therefore, it is important to design and fabricate fast-charging anodes with high-energy density.

To address these problems, we propose a novel Si–Graphite composite design, which not only possesses the enhanced kinetics for Li$^+$ but also satisfies the aforementioned industrial electrode conditions. We prepare a composite consisting of an edge-plane-activated graphite and a-Si nanolayer (SEAG) through nickel-catalyzed hydrogenation (Ni + C$_{graphite}$ + 2H$_2$ → Ni + CH$_4$)[27, 28] and chemical vapor deposition (CVD) using both acetylene (C$_2$H$_2$) and silane (SiH$_4$) gas. In contrast to earlier approaches, the material design in the current work provided multiple attractive advantages. First, the catalytic reaction primarily activates the Li$^+$-reactive edge plane of graphite[28–31]. According to previous studies, the mass-transfer kinetics of graphite can be improved by creating exposed edge sites[32–34]. As a result, the kinetics is enhanced with the minimized surface area, leading to a high initial CE. Second, the remaining Ni nanoparticles, which act as a catalyst for the activation reaction, improve the electric conductivity of SEAG[35]. Third, despite the catalytic hydrogenation, the graphite core still remains as a supporting framework that withstands high mechanical pressure during electrode calendering[36]. Besides, since the dense core may have sustained the tap density of the composite[24, 37], such features are more favorable to the attainment of high electrode density. Fourth, the nanoscale Si coating layer increases the energy density of the material and allows fast Li diffusion owing to its high specific capacity, and shortened the Li$^+$ diffusion length[38, 39]. Finally, all the synthesis methods consist of simple processes. In particular, the heat treatment, including the catalytic hydrogenation, and CVD procedures can be conducted as a continuous process in the same furnace. Together, these advantages strengthen the commercial feasibility of SEAG in fast-charging high-energy LIBs.

## Results

**Materials synthesis and structural design.** The schematic in Fig. 1a illustrates the procedures for the fabrication of SEAG composite. Mesocarbon microbeads (MCMBs), which are widely adopted as a commercial anode with high tap density (1.38 g cm$^{-3}$) and excellent CE in the 1st cycle (95.7%), were used as pristine graphite. At first, spherical nickel nanoparticles with a size of ~500 nm were formed on the pristine graphite via a simple reflux method at 80 °C. The sample was then calcined at 1000 °C in a hydrogen (H$_2$) atmosphere to trigger the catalytic hydrogenation reaction between nickel and graphite. As carbon atoms were hydrogenated, the adsorbed Ni nanoparticles penetrated the graphite core with methane (CH$_4$) gas evolution, resulting in a holey structure. Note that it was thermodynamically more favorable for the catalytic gasification of carbon to take place on the edge plane than on the basal plane of graphite[27–31]. As a final step, a graphitic carbon shell and an amorphous Si (a-Si) nanolayer were homogeneously distributed on nickel and graphite, respectively, via consecutive CVD processes using C$_2$H$_2$ and SiH$_4$ gases (details are given in the "Methods" section). Additionally, the elemental composition of SEAG was analyzed by various kinds of methods, including inductively coupled plasma optical emission spectrometry, thermogravimetric analysis, and elemental analyzer (Supplementary Fig. 1 and Supplementary Table 1).

For the detailed elucidation of its unique structural characteristics, the cross-section of SEAG is schematically shown in Fig. 1b. The Ni nanoparticle pierces the graphite along the edge plane, while it functions as a catalyst for the gasification of carbon. A high-resolution transmission electron microscope (HR-TEM) image of the embedded Ni is shown in Fig. 1c. Owing to the catalytic reaction, unrevealed edge planes emerged, and these activated edges improved the mass-transfer property of Li$^+$ ions with enlarged electrochemically active sites. To demonstrate the enhanced kinetics for Li$^+$, we conducted cyclic voltammetry and estimated the relative electrochemically active surface area of the edge-activated graphite (EAG) from the relationship between the peak current and scan rate, described by the Randles–Sevcik equation (Fig. 1d and Supplementary Fig. 2)[40]. According to this equation, the slope of the plot in Fig. 1d is proportional to the active surface area (details are described in Supplementary Note 1). As a result, the Li$^+$-reactive surface area in EAG was about twice that of pristine graphite. Furthermore, despite the activation process, SEAG still had the inner core as a ductile framework, which enabled it to be easily calendered to yield a high electrode density.

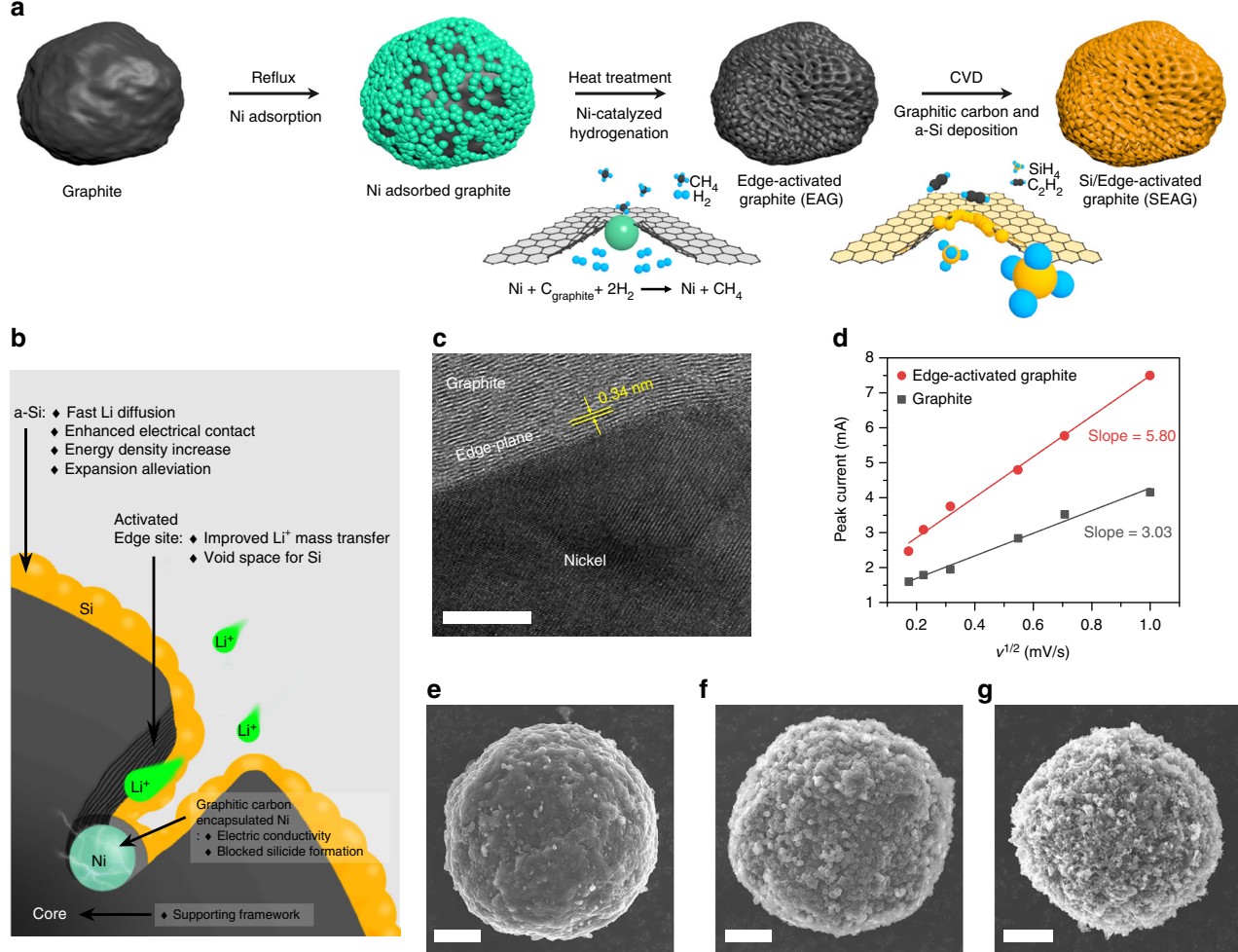

**Fig. 1** Schematic of the procedures for fabrication and characterization of SEAG. **a** Fabrication of SEAG: the adsorbed Ni penetrated graphite via catalytic hydrogenation at 1000 °C, which led to edge-plane activation on the surface of graphite. **b** Cross-sectional illustration showing the detailed structural characteristics of SEAG. **c** HR-TEM image of a Ni nanoparticle embedded in EAG. **d** Relationship between the square root of the scan rate and the peak current: the slope indicates the relative electrochemically active surface area. SEM images of pristine graphite **e**, graphite with adsorbed Ni on its surface **f**, and the SEAG composite **g**. Scale bars, 5 nm **c**, 5 μm **e**, **f**, **g**

The alloying reaction of the remaining Ni nanoparticles in EAG into Ni silicide inevitably occurred under the $SiH_4$ atmosphere, before the gas decomposition temperature was reached, because of the high reactivity with Si (Supplementary Fig. 3)[41–43]. Unfortunately, this alloying reaction had critical influences on the electrochemical performance, such as decreasing the gravimetric capacity by Si consumption, reducing the electrical conductivity of Ni[44], and causing an irreversible capacity with a poor initial CE[45]. Thus, a well-defined barrier against silicide formation was necessary to secure good battery performance. Hence, the CVD processes using $C_2H_2$ and $SiH_4$ were successively employed to develop the protective carbon shell on Ni, followed by a Si coating layer. Because of the high carbon solubility of nickel[46], the electrically conductive graphitic shell was deposited mainly on the nickel nanoparticles, and then the Si layers were uniformly formed on both the surface of this shell and graphite without the formation of Ni silicide. The homogeneous a-Si nanolayer led to fast $Li^+$ diffusion and alleviated volume expansion while attaining high-energy density. These Si layers synthesized by the CVD method could establish good electrical contact with graphite by adhering well onto the graphite surface[38]. Scanning electron microscope (SEM) images of pristine graphite, graphite with adsorbed Ni, and SEAG are shown in Fig. 1e–g, respectively.

**Physical characterization**. We carried out X-ray diffraction (XRD) measurements, the analysis for the particle size distribution and tap density, and Brunauer–Emmett–Teller (BET) measurements to evaluate the physical properties of SEAG (Fig. 2). The characteristic peaks in the XRD pattern (Fig. 2a) indicate the existence of graphite; however, Si peaks are absent because the Si layer on the SEAG was amorphous[47]. The average diameter of the SEAG particles was 22 μm, which is almost the same as that of pristine graphite (Fig. 2b). Moreover, the tap density was measured as 1.27 g cm$^{-3}$ in SEAG, which is comparable to that of pristine graphite but higher than that of conventional natural graphite (NG) (Fig. 2c). The BET-specific surface area of SEAG was estimated to be 2.49 m$^2$ g$^{-1}$ (Fig. 2d), which is slightly higher than that of pristine graphite, but still in the range obtained for conventional NG. Since irregular particle-size distribution, low tap density, and excessively large surface area are unfavorable for electrode preparation owing to inhomogeneous slurry mixing[48], SEAG is expected to be advantageous for conventional electrode fabrication.

**Structural characterization of SEAG composite**. We performed SEM analysis with a focused ion beam (FIB) and HR-TEM analysis with a high-angle annular dark field scanning

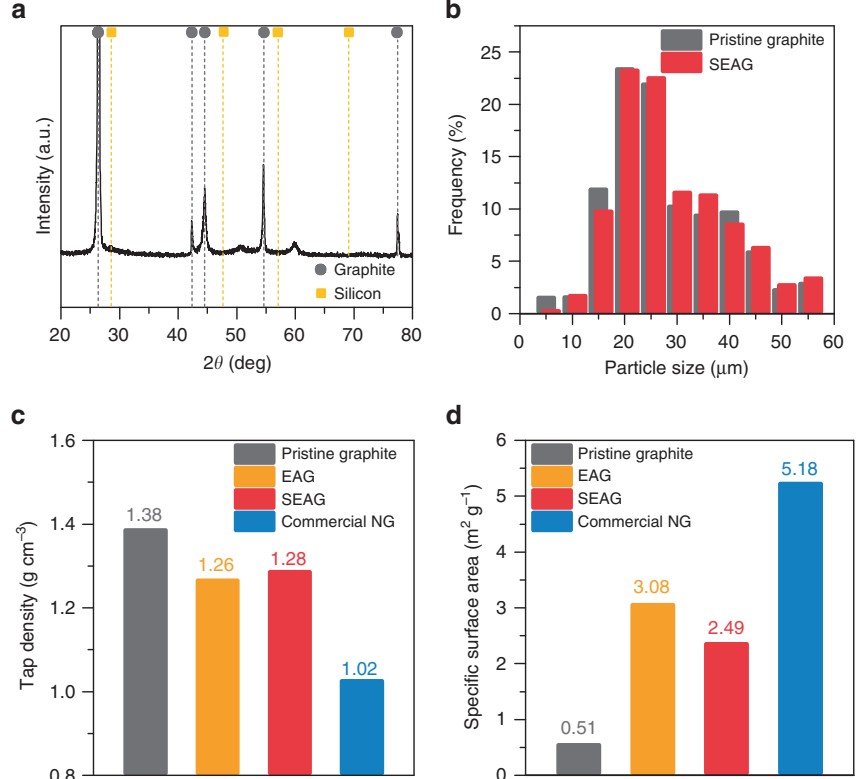

**Fig. 2** Physical properties of SEAG. **a** XRD pattern of SEAG. **b** Statistical analysis of the particle-size distribution of pristine graphite (gray) and SEAG (red). **c** Tap density and **d** specific surface area of pristine graphite (gray), EAG (orange), SEAG (red), and conventional NG (blue)

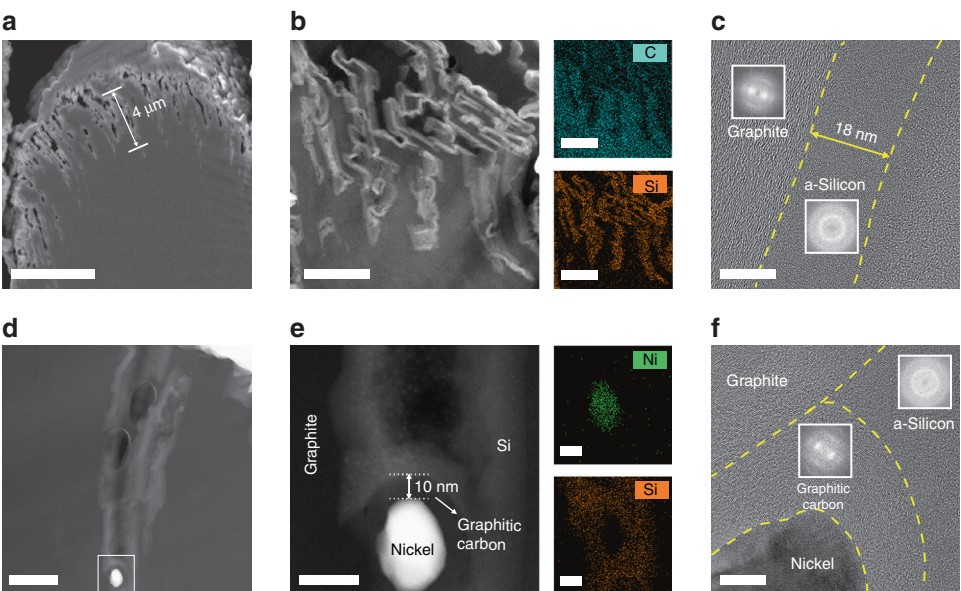

**Fig. 3** Detailed investigation of cross-sectional SEAG. **a** SEM image of SEAG in cross-sectional view: SEAG maintained a rigid inner graphite framework despite going through catalytic hydrogenation. HAADF-STEM images with EDS mapping analysis of **b** the activated holes and silicon nanolayer on the graphite surface and **d** the nickel nanoparticle piercing the core of graphite. **e** Magnified HAADF-STEM image of **d**. HR-TEM images at the interfacial region of **c** SEAG and **f** a nickel nanoparticle; fast Fourier-transform images are shown in the insets. The nickel nanoparticle was clearly separated from silicon by the graphitic carbon shell, which prevented the unfavorable formation of Ni silicide. Scale bars, **a** 5 μm, **b** 1 μm, **c**, **f** 10 nm, **d** 200 nm, **e** 30 nm

transmission electron microscope (HAADF-STEM) for detailed characterization of SEAG (Fig. 3). As shown in the cross-sectional SEM image (Fig. 3a), SEAG retained the sturdy inner graphite framework that withstands mechanical pressure even though it underwent catalytic hydrogenation, which is advantageous for

increasing the electrode density. The HAADF-STEM image shows the magnified cross-sectional morphology of the SEAG (Fig. 3b). Contrary to the dense interior of pristine graphite (Supplementary Fig. 4), the micrometer-sized holes (average depth of 4 μm) in SEAG were well arranged on the surface in one

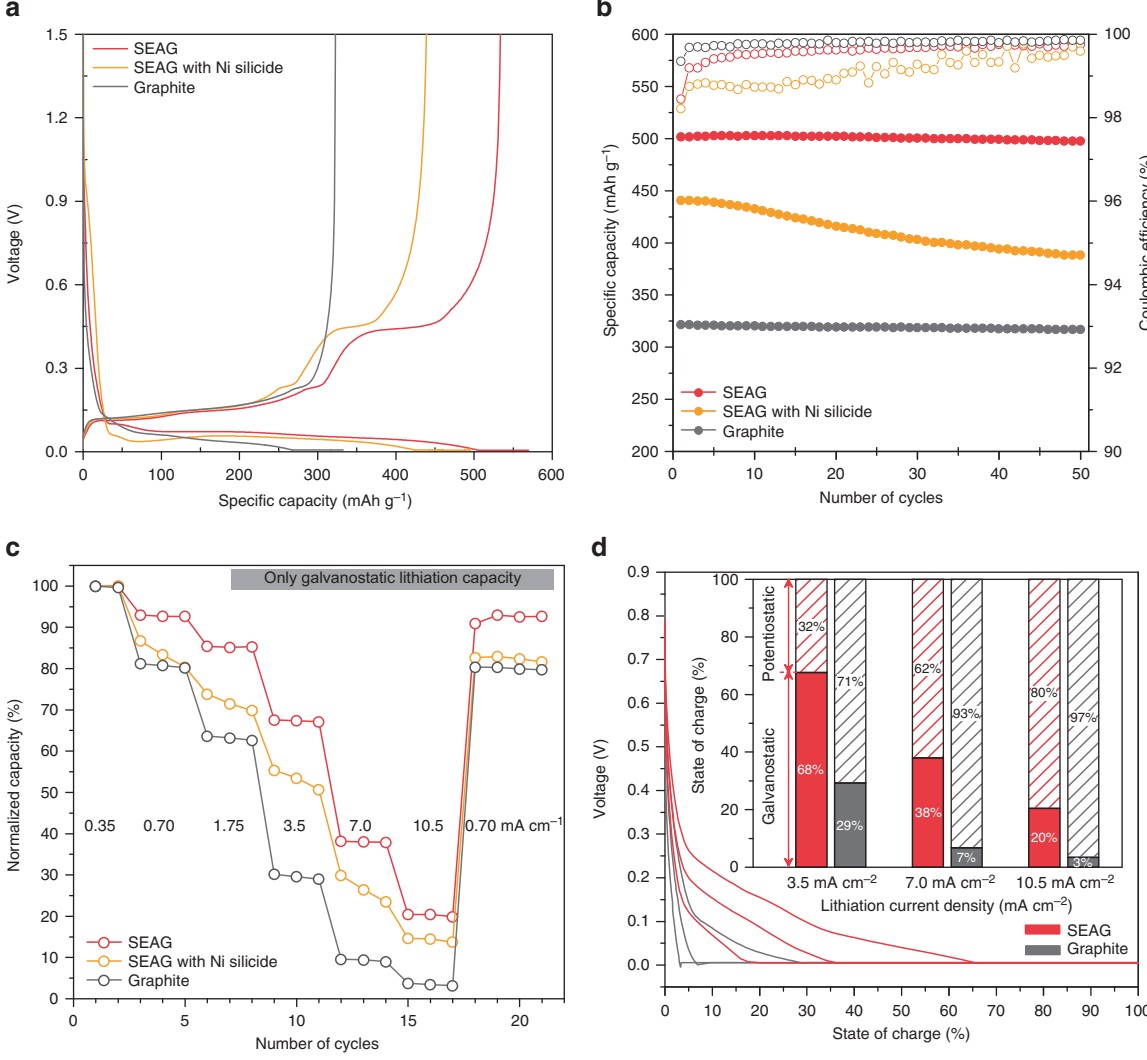

**Fig. 4** Electrochemical characterization of various anodes in half-cell configurations. **a** Voltage profiles of SEAG, SEAG with Ni silicide, and graphite in the 1st cycle. **b** Plots of reversible capacity and cycling CE vs. cycle number for SEAG, SEAG with Ni silicide, and graphite over 50 cycles. The **a** 1st cycle and **b** cycling test were carried out at current densities of 0.35 and 1.75 mA cm$^{-2}$, respectively (1C = 3.5 mA cm$^{-2}$). **c** Galvanostatic charge capacities of SEAG under various lithiation current densities from 0.35 to 10.5 mA cm$^{-2}$, compared to SEAG with Ni silicide and graphite. **d** Voltage profiles during charging process of SEAG and graphite, measured at increasing current densities from 3.5 to 10.5 mA cm$^{-2}$; the inset shows a plot of SOC divided into galvanostatic/potentiostatic stages at each current density

side of the particle, and the Si layer coated the inner holes as well as the outer surface of graphite. It was clarified via energy-dispersed X-ray spectroscopy (EDS) mappings of elemental carbon (cyan) and Si (orange) in SEAG (Fig. 3b). In addition, the HR-TEM image with fast Fourier transform analysis shows that the a-Si layer with thickness of ~18 nm coated the graphite surface (Fig. 3c). Such incorporation of nanoscale a-Si in graphite could alleviate the stress and strain induced by the volume expansion and allow fast Li$^+$ diffusion[38, 49, 50]. X-ray photoelectron spectra (XPS) indicates the presence of a thin oxide layer on the a-Si surface due to the air exposure (Supplementary Fig. 5).

As shown in Fig. 3d and e, a Ni nanoparticle that pierced the graphite core is completely enclosed within the graphitic carbon shell, leaving no space between Ni and carbon. Even though this graphitic carbon shell was only 10-nm thick, it completely covered the entire Ni nanoparticle and blocked the penetration of SiH$_4$ during the decomposition process. As shown by the EDS mapping analysis (Fig. 3e), elemental Ni (green) and Si (orange) are apparently separated from each other in the SEAG particle.

The HR-TEM image (Fig. 3f) and XRD analysis (Supplementary Fig. 6) clearly confirm the segregated existence of Ni, graphitic carbon, and a-Si. Unlike amorphous carbon, the graphitic carbon provided high electrical conductivity[51] and minimized the irreversible capacity loss caused by Li$^+$ trapping[52]. In short, the graphitic carbon shell could help address the crucial problems resulting from the formation of Ni silicide by efficiently blocking the penetration of SiH$_4$. In addition, it enhanced the electrochemical properties of SEAG with its high electrical conductivity and low trapping of Li$^+$.

**Electrochemical performance of SEAG composite in half-cell.** The electrochemical properties of SEAG were evaluated with a coin-type half-cell at 25 °C to investigate the favorable effects of SEAG design on battery performance, especially the capability for fast charging (Fig. 4). All electrodes were fabricated with high areal capacity (3.5 mAh cm$^{-2}$), high electrode density (1.6 g cm$^{-3}$, ~33% of electrode porosity), and minimum use of binder

materials (1.5 wt% of carboxymethyl cellulose and styrene buta-diene rubber (SBR)) and conductive agent (1.0 wt% of carbon black). Even under these challenging electrode conditions, the SEAG achieved an outstanding CE (93.8%) with high specific capacity (525 mAh g$^{-1}$) in the 1st cycle (Fig. 4a). To the extent of our knowledge, this is one of the highest values for initial CE among the previously reported graphite–Si composites. Accord-ing to the differential capacity plot in Supplementary Fig. 7, it is clear that its higher specific capacity is owing to the contribution of Si nanolayer, and the total capacity of SEAG coincides well with the calculated value based on ICP-OES result (Supplemen-tary Note 2). In addition, SEAG exhibited superior cycling sta-bility for 50 cycles (Fig. 4b) with 99.3% capacity retention at a current density of 1.75 mA cm$^{-2}$ (extended cycle performance and magnified plot of cycling efficiency are presented in Sup-plementary Figs. 8, 9). The outstanding electrochemical perfor-mances of SEAG are attributed to the uniform a-Si nanolayers and the enhanced electrical conductivity derived from the gra-phitic carbon capped with Ni nanoparticles. The post-cycling TEM analysis was conducted to verify the structural stability of SEAG during cycles (Supplementary Fig. 10). Furthermore, the effect of electrolyte additives on the cycle performance of SEAG was investigated and the result is shown in Supplementary Fig. 11. SEAG with Ni silicide exhibited an inferior electro-chemical performance that included lower specific capacity (435 mAh g$^{-1}$), poor CE in the 1st cycle (88.5%), and severe capacity fading for 50 cycles (88% of initial capacity), indicating that blocking the formation of Ni silicide is of great importance for better performance.

To examine the lithiation behavior, which is highly affected by the over-potential, the rate capability of each sample was measured by only the galvanostatic method with lithiation current densities of 0.7–10.5 mA cm$^{-2}$ (Fig. 4c). At higher applied currents, the lithiation capacity of graphite severely deteriorated, and exhibited only about 2% of the initial capacity (8 mAh g$^{-1}$) at a high current density of 10.5 mA cm$^{-2}$. On the other hand, EAG demonstrated improved lithiation behavior with high lithiation capacity at a current density of 7 mA cm$^{-2}$, which was over twice that of graphite (Supplementary Fig. 12). In addition, such EAG electrode exhibited better fast-charging performance than that of graphite electrode containing 5 wt% of carbon black (Supple-mentary Fig. 13). These improvements suggest that the activation of unrevealed edge plane and embedded Ni nanoparticles could enhance both the mass and charge transfer kinetics during lithium intercalation. To the best of our knowledge, considering the electrode conditions, the lithiation rate capability of EAG in this study is the best performance reported among the graphite anodes (Supplementary Table 2).

Moreover, the rate property of SEAG was further improved with the incorporation of a-Si nanolayer. A specific lithiation capacity was obtained at 10.5 mA cm$^{-2}$, which is almost 20% of the initial capacity (100 mAh g$^{-1}$) and is 10 times the value obtained for graphite. This remarkable improvement supports our hypothesis that a homogeneous a-Si nanolayer allows high reactivity and fast diffusion for Li-ion transport[49, 50]. Additional-ly, the differential capacity plot clearly shows that the enhanced performance of SEAG is owing to the unrevealed edge plane and a-Si nanolayer (Supplementary Fig. 14).

The fully lithiation properties, which were determined via both galvanostatic and potentiostatic methods, were further investi-gated in detail (Fig. 4d). The results are consistent with the previous rate properties estimated using only the galvanostatic stage: while SEAG showed a relatively mitigated voltage drop at each current density, graphite rapidly reached the cut-off voltage with severe over-potential, which is considered the main reason for both the long potentiostatic stage and Li-plating phenomenon.

As seen in the inset of Fig. 4d, most of the lithiation capacity (93 and 97% at 7.0 and 10.5 mAh cm$^{-2}$, respectively) of graphite originated at the potentiostatic stage and almost 119 min were needed for fully lithiation at 10.5 mAh cm$^{-2}$ (Supplementary Fig. 15). In contrast, a relatively large portion of lithiation capacity of SEAG was derived from the galvanostatic region with shorter charging time (66 min at 10.5 mA cm$^{-2}$) because of its alleviated over-potential. It should be noted that it was difficult to control the charging time in the potentiostatic method because the current density became variable, and thus attaining fully lithiation on time was impeded by the prolonged potentiostatic process, even under high charging current density.

**Fast-charging performance and volumetric energy in full-cell.** To demonstrate the viability of using SEAG in practical appli-cations, we performed a pouch-type full-cell evaluation with high-voltage LiCoO$_2$ (LCO) as a cathode in the voltage range of 2.7–4.35 V (Fig. 5). The electrochemical performance of LCO in the half-cell is presented in Supplementary Fig. 16. The capacity ratio of negative to positive electrode (N/P ratio) was fixed at 1:1, with an areal cathode capacity of 3.4 mAh cm$^{-2}$. At the formation cycle (at 0.34 mA cm$^{-2}$), SEAG exhibited a high discharge capa-city of 3.25 mAh cm$^{-2}$ with a high CE of 92%, which are com-parable with the values obtained for graphite (3.33 mAh cm$^{-2}$ with 94% CE) (Supplementary Fig. 17). Also, SEAG demonstrated a competitive cycle stability at the current density of 1.7 mA cm$^{-2}$ compared to that of pristine graphite (Supplementary Fig. 18). Moreover, the cycling tests were conducted with various charging current densities (5.1, 7.7, and 10.2 mA cm$^{-2}$) to investigate the fast-charging capability of each sample. The 1st voltage profiles at each current (Fig. 5a–c) clearly show that the SEAG occupies a lower over-potential with a shorter potentiostatic region than that of graphite. This result agrees well with the half-cell data pre-sented in Fig. 4.

The cycling tests showed that the SEAG demonstrated better capacity retention over 50 cycles at all the current densities, whereas graphite exhibited drastic capacity fading after just 10 cycles (Fig. 5d–f). In the Coulombic efficiency plot, SEAG rapidly achieved a stabilized efficiency of > 97% at the 1st CE, regardless of the applied charging current. On the other hand, graphite suffered from extremely unstable efficiency with increasing current density, especially during 10 cycles, with a low 1st cycle efficiency of 92, 86, and 83% at each current density. These severe energy losses with the inferior CE are believed to be the result of irreversible Li metal plating on the electrode surface[5–11]. The overall voltage profiles of the cycling tests are presented in Supplementary Fig. 19.

In addition, we plotted the time required to charge to 80% of the state of charge (SOC) as applied charge current density (Fig. 5g). The SEAG electrode reached the SOC faster than graphite at all charging rates; the charging time of SEAG was much shorter (by a factor of ~1.5) because of its lower over-potential. Furthermore, the energy density is plotted as a function of charging current density in Fig. 5h (details of the measure-ments are given in Supplementary Table 3). As the applied current density was increased, a significant energy loss was observed at the graphite electrode (726 Wh l$^{-1}$ at 10.2 mA cm$^{-2}$). However, the SEAG electrode retained the highest volumetric energy even under a high charging current density (1060 Wh l$^{-1}$ at 10.2 mA cm$^{-2}$).

**Discussion**
To interpret the result of the fast-charging cycling tests, the irreversible change in the electrode thickness was analyzed via dilatometry in the full-cell (Fig. 6a). Liu et al. reviewed the adverse

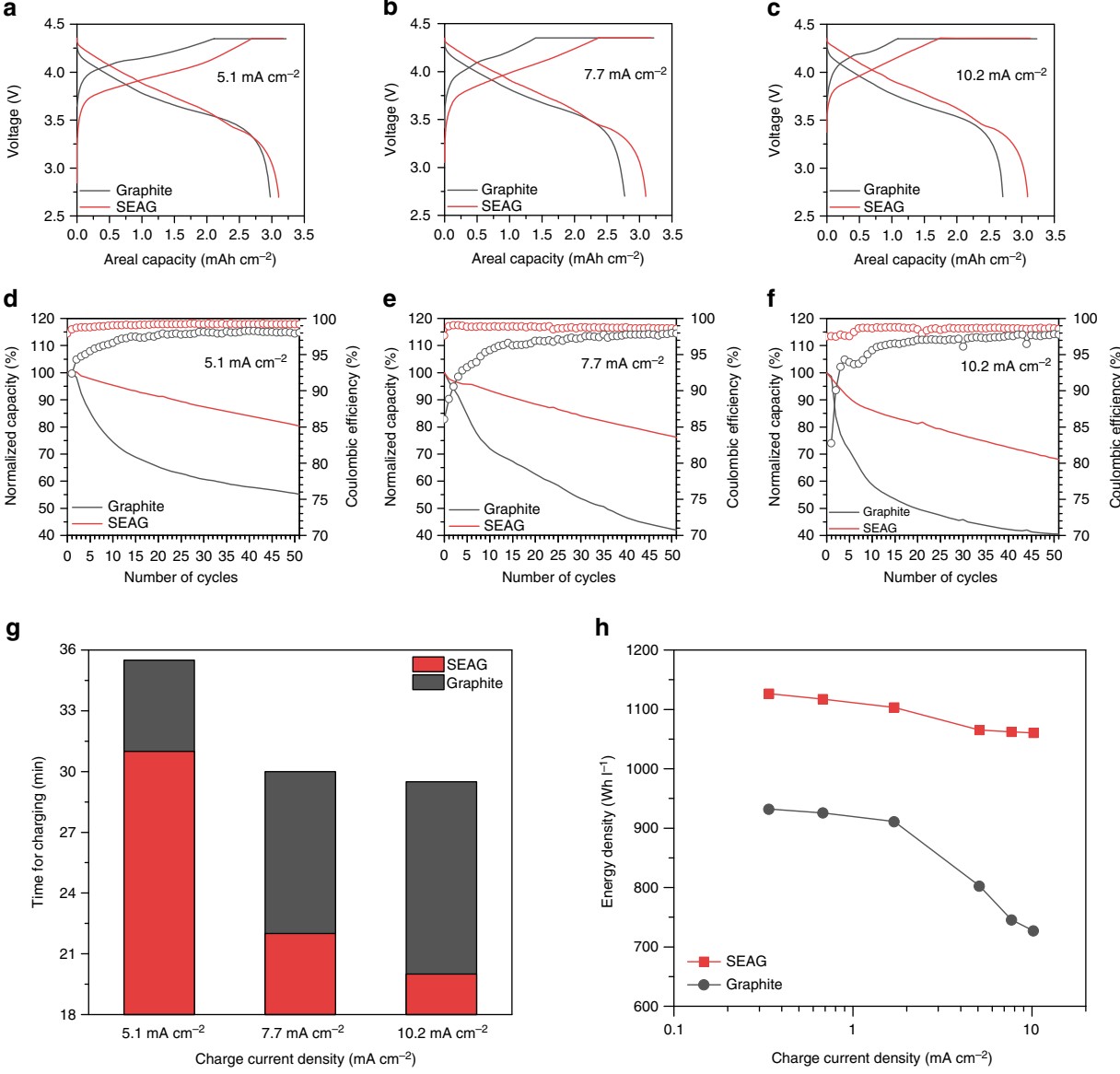

**Fig. 5** Fast-charging performance of full-cells with anodes of graphite and the SEAG composite. Voltage profiles at charging current density of **a** 5.1 mA cm$^{-2}$, **b** 7.7 mA cm$^{-2}$, and **c** 10.2 mA cm$^{-2}$. Fast-charging cycling performance in the potential range from 4.35 to 2.7 V for 50 cycles under increasing charging current densities of **d** 5.1 mA cm$^{-2}$, **e** 7.7 mA cm$^{-2}$, and **f** 10.2 mA cm$^{-2}$. The discharging current density was fixed at 1.7 mA cm$^{-2}$ in all cycling tests. **g** Time required to charge to 80% of SOC at each current density. **h** Plot of volumetric energy density vs. applied charging current density for graphite and SEAG. The volumetric energy densities were calculated from the total thickness of both the cathode and anode, considering the electrode volume expansion at the lithiated state during the first cycle. In the electrode composed of SEAG, a volumetric energy density of 1060 Wh l$^{-1}$ was delivered under an applied charging current density of 10.2 mA cm$^{-2}$

effects of lithium plating, which brought about a drastic energy loss and raised the internal resistance of the cell; these results originated from the irreversible consumption of Li$^+$ ions and a thickened solid electrolyte interphase layer that involved the depletion of electrolyte[5]. In addition, the continuous growth of metallic lithium induces an internal short circuit, which is considered a safety issue[8]. Therefore, Li plating is regarded as a critical index for battery degradation and as a guide to understanding the rapid energy fading and unstable efficiency, especially under a high charging current density. In this study, we characterized the lithium plating through irreversible dilation, which was estimated by measuring the difference in cell thickness in the delithiated state at high and standard charging rates (7.7 and 1.7 mA cm$^{-2}$, respectively)[53].

Surprisingly, a much smaller thickness change was measured in SEAG when compared with that in graphite, even though Si, which causes large volume expansion, was incorporated into the SEAG. The rapid and continuous increase in the dilation of graphite, which was the result of metallic lithium deposition, signified the drastic capacity degradation. Photographs and SEM images of the electrode that was subjected to high charging current density (7.7 mA cm$^{-2}$) for 50 cycles are shown in Fig. 6b–g. The electrode composed of SEAG showed a relatively clear surface without any trace of lithium deposition (Fig. 6b–d). In contrast, a considerable amount of metallic lithium can be observed on the surface of the graphite electrode; the lithium passivated the electrode to reach a thickness of 30 μm, leading to severe energy loss and irreversible dilation (Fig. 6e–g).

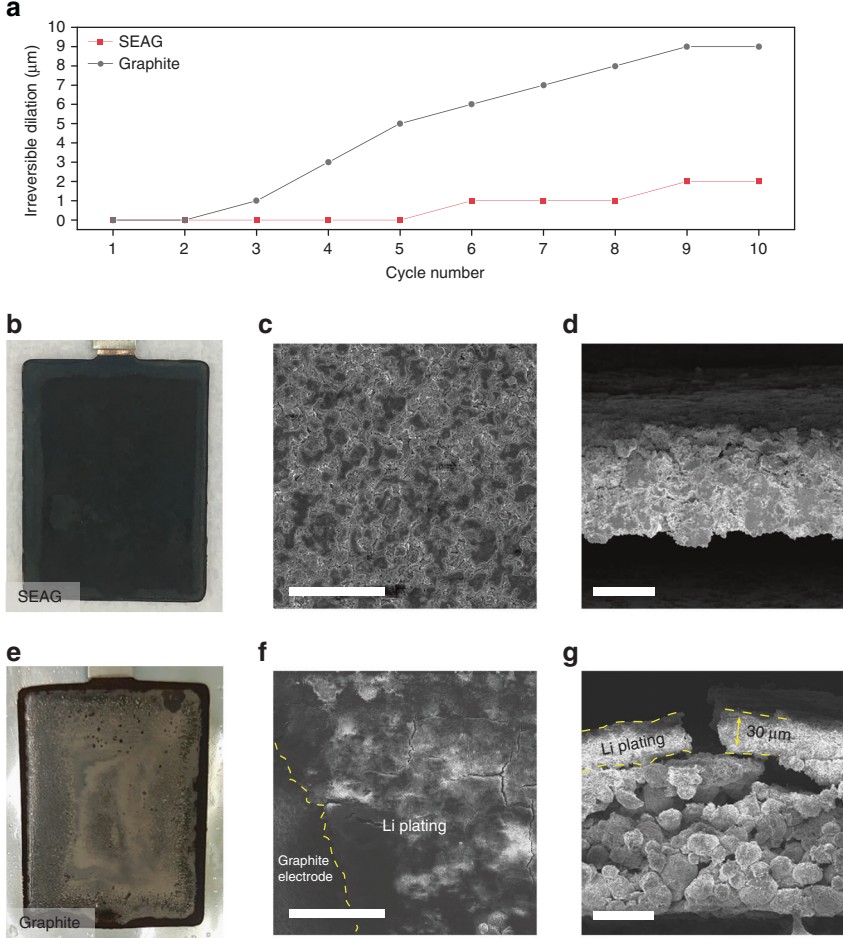

**Fig. 6** Dimensional and morphological changes of the electrodes after fast-charging cycles. **a** Irreversible increase in thickness of the electrode at the delithiated state under 7.7 mA cm$^{-2}$ for 10 cycles. Photographs of **b** SEAG electrode and **e** graphite electrode after 50 cycles at 7.7 mA cm$^{-2}$. Top view and cross-sectional SEM images of **c**, **d** SEAG electrode and **f**, **g** graphite electrode. Yellow dash line indicates the deposited Li, which reacted with the electrolyte. Scale bars, **c** 100 μm, **d**, **g** 50 μm, **f** 400 μm

In summary, an elaborate SEAG composite, consisting of a-Si nanolayer and edge-plane-activated graphite, was prepared via a simple synthetic route to realize a fast-charging high-energy anode. With this well-developed SEAG, the electrochemical performance of both a half-cell and full-cell were evaluated even under industrial electrode conditions such as high areal capacity loading (≥3.4 mAh cm$^{-2}$) and high electrode density (~1.6 g cm$^{-3}$) with a limited content of conducting agents (≥1 wt%). In the half-cell, the SEAG achieved a high specific capacity (525 mAh g$^{-1}$) with remarkable CE (93.8%) in the 1st cycle and good capacity retention (99.3%) for 50 cycles. Moreover, at a high charging current density (10.5 mA cm$^{-2}$), the lithiation capacity of SEAG (with relatively stable voltage behavior) was ten times that of the conventional graphite. In the full-cell with high-voltage LCO, the SEAG electrode exhibited enhanced fast charge performance with mitigated polarization and rapidly stabilized CE, leading to a higher volumetric energy density (1060 Wh l$^{-1}$) with 1.5 times shorter charging time than that of conventional graphite even at a harsh charging current of 10.2 mA cm$^{-2}$. Furthermore, despite the Si content in SEAG, a much smaller irreversible increase in the electrode thickness (2 μm) was observed than in the graphite (9 μm), without any trace of lithium plating at a high charging current density (7.7 mA cm$^{-2}$). Concomitantly, we believe that the fast-charging properties can be further improved by increasing the ionic and electric conductivities with electrolyte modifications and advancement in cathode materials. Such

excellent electrochemical performance of SEAG clearly demonstrates its viability in fast rechargeable high-energy battery applications.

## Methods

**Synthesis of SEAG.** For the Ni adsorption on the graphite, 50 g of pristine graphite (MCMB, Osaka gas), 6 g of nickel chloride hexahydrate (NiCl$_2$· 6H$_2$O, >97.0%, JUNSEI), and 0.2 g of sodium hydroxide (anhydrous NaOH, >98.0%, bead form, SAMCHUN) were dissolved in methanol (>99.5%, SAMCHUN)/deionized water (5:5, v/v), followed by the addition of 1 ml of hydrazine monohydrate (N$_2$H$_4$·H$_2$O, 98.0%, Sigma-Aldrich). The solution was heated at 78 °C for 30 min in air atmosphere by reflux technique. The Ni adsorbed graphite was obtained through the centrifugation. In order to trigger the catalytic hydrogenation, the prepared samples were annealing in the furnace at 1000 °C for 3 h under H$_2$ (99.999%, KOSEM) atmosphere (1000 sccm). In succession, for the formation of both the graphitic carbon shell and a-Si nanolayer on the graphite, C$_2$H$_2$ gas (10.0%, N$_2$ balance, KOSEM) was flowed at 900 °C for 10 min (1000 sccm) and then SiH$_4$ gas (99.9999%, KOSEM) was introduced into the furnace at 500 °C for 30 min (50 sccm). In case of SEAG with Ni silicide, the process of C$_2$H$_2$ flow was omitted.

**Materials characterization.** Structural characterization of the samples was carried out using SEM (Verios 460, FEI). Dual beam FIB (Helios 450HP, FEI) was performed for scanning the particles in the cross-sectional view. HR-TEM (JEM-2100F, FEI) was conducted for detailed analysis. EDS was utilized in SEM (EDS, XFlash 6130, Bruker) and in HR-TEM (EDS, Aztec, Oxford). The oxide layer on a-Si surface was analyzed by XPS (K-alpha, Thermo Fisher). XRD (D/Max2000, Rigaku) was carried out for the powder analysis using Cu-Ka radiation, a scan range of 20°–80°, a step size of 0.02°, and a counting time of 5 s. Particle size distribution was measured by the Fraunhofer approximation by laser diffraction

particle size analysis instrument (Microtrac S3500, Microtrac). Tap density was determined by density analyzer (GeoPyc 1360, micromeritics). Specific surface area was estimated with the BET theory with porosity and surface area analyzer (TriStar II, micromeritics).

**Electrochemical characterization.** For fabrication of the working electrode, the slurry composed of the active material, the conductive agent (Super P, Timcal), and the binder materials (sodium carboxymethyl cellulose (CMC, Nippon paper) and SBR (Zeon)) was uniformly mixed by homogenizer in the mass ratio of 96:1:1.5:1.5 and casted onto the copper foil up to 3.5 mAh cm$^{-2}$ of areal capacity loading. In sequence, the electrode was dried at 80 °C and calendared for 1.6 g cm$^{-3}$ of electrode density by roll press. The electrode underwent vacuum drying at 110 °C for 12 h. In order to assemble the cell, CR2032 (half-cell) and pouch (full-cell) type cell were utilized in argon-filled glove box and dry room, respectively. In case of half-cell, lithium metal (>99%, Honjo metal) was used as counter electrode. The electrolyte was 1.3 M LiPF$_6$ in mixture of ethylene carbonate/ethyl methyl carbonate/diethyl carbonate (3/5/2, by volume) with 10% of fluoroethylene carbonate, 0.2% of lithium tetrafluoroborate, 0.5% of vinylene carbonate, 3% of succinonitrile, and 1% of propane sultone (Panax Starlyte). As a separator, microporous polyethylene (15 µm, Celgard) was used. Electrochemical properties of the half-cell were estimated under the potential window from 0.005 to 1.5 V for the first cycle, and from 0.005 to 1.0 V for the rest of cycles. In full-cell, single-crystal lithium cobalt oxide (LCO, homemade) was adopted as cathode electrode with 1:1 of N/P ratio. The cathode electrode was fabricated with active material, carbon black, and polyvinylidene fluoriade binder (PvdF, Solef) in mass ratio of 96:2:2. The mass loading level of the cathode was 20 mg cm$^{-2}$ and the electrode density was tuned up to 3.6 g cm$^{-3}$. Electrochemical tests of the full-cell were performed in the voltage window between 2.7 and 4.35 V. All electrochemical tests were conducted using a battery cycler (TOSCAT-3100, Toyo system).

**Dilatometry.** The irreversible volume expansion of the electrode was measured at delithiated state as the thickness difference between in case of high charging rate (7.7 mA cm$^{-2}$) and standard charging rate (1.7 mA cm$^{-2}$). The discharging current density was fixed as 1.7 mA cm$^{-2}$ in both cases. Thickness change was estimated by dilatometer (Mitutoyo).

**Data availability.** The authors declare that the data supporting the findings of this study are available within the article and its Supplementary Information files. All other relevant data supporting the findings of this study are available on request.

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

## Acknowledgements

This work was supported by the project from Samsung Research Funding & Incubation Center of Samsung Electronics Co. Ltd (A Rapid Charging Battery and its Prototype Cells Retaining High Energy Density, SRFC-TA1603-01).

## Author contributions

N.K. and M.K. conceived and designed the experiments. N.K. prepared the samples and carried out all the main experiments. S.C. assisted with the data analysis and conducted dilatometry; J.M. assisted with sample preparation. N.K. and J.C. co-wrote the paper. N.K., S.C., M.K., and J.C. discussed the results and revised or commented the manuscript.

## Additional information

**Competing interests:** The authors declare no competing financial interests.

