## [Peer Review File · Nature Communications]

Reviewers' comments:

Reviewer #1 (Remarks to the Author):

This paper reports a novel method for obtaining increased rate capability in graphite anodes. The approach is interesting, and the results shown here demonstrate that it is viable. I have several relatively minor criticisms:

(1) The rationale for studying these materials is not particularly well written. Here, the authors should probably employ the services of a native English speaker.

(2) In their introduction, they do not say enough about other attempts to obtain faster kinetics by exposing graphite edges. A couple of references are mentioned (e.g., the engineered porosity in ref 12), but they should really discuss the idea of creating exposed edges here in more detail, since this is central to their work. Relevant prior studies include: Persson et al, J Phys Chem (2010), 1, 1176 and Mukhopadhyay et al, Adv. Functional Mater (2013), 23, 2397 (there is other pertinent work as well).

(3) In several places, they emphasize the need for commercially viable approaches, yet the CVD approach seems potentially costly.

(4) The authors do not really discuss SEI effects. Their surface treatment almost certainly alters the SEI, and this could be relevant. For example, the FEC added to the electrolyte is known to improve performance with Si containing electrodes, and this could be important. Have the authors looked at the behavior of their materials with any other electrolytes, to see if the improvements dependent on the electrolyte ?

(5) The a-Si layer could slow down diffusion. Since this layer is thin, it does not appear to be a limiting factor (based on their results). However, this should be discussed more clearly.

(6) An estimate of the amount of Si added to the electrodes would be useful. A very thin Si layer on a dense particle of this size would presumably not provide this much increased capacity ? The porosity of the graphite is presumably important here ?

(7) Given the large expansion and contraction of the Si during cycling, it's a little surprising that the material still functions reasonably well after 50 cycles. Did the authors try to do any post-cycling TEM (this is admittedly challenging) ?

Reviewer #2 (Remarks to the Author):

This work reported a high-performance anode material based on a-Si nanolayer deposited on edge-site activated graphite (SEAG). This anode material, SEAG, showed an excellent initial coulombic efficiency (~94 %), a large areal capacity (~ 3.5 mAh cm⁻²) and a high electrode density (~ 1.6 g cm⁻³). All these results are highly desirable, so very attractive for the industrial application. But some data could not support the results well.

1) How about the output of one batch (g or Kg) and homogeneity of the product? CVD is an effective technique to deposit a layer on the surface of another material, but the homogeneity of as-obtain

product is a great challenge. Because the gas penetration into the powders is limited, the particles nearly below or around the surface might be well coated. But those at the bottom have less chance to be coated. So, is there any solution to address this issue?

2) Since a-Si is directly exposed to environment, it would be heavily oxidized even at room temperature. Then, the resultant SiO_x would greatly reduce the initial coulombic efficiency, which seems to be opposite to the high value reported here (~94 %). So, please check the XPS spectra and discuss this in detail.

3) The fast charging of SEAG is not so good as the authors acclaimed in the manuscript. In view of the high areal capacity (~ 3.5 mAh cm⁻²), the rate capability up to 10.5 mA cm⁻² (~ 3 C) is not enough. Furthermore, the capacity retention dropped to 20 % of the initial capacity. These results indicate that the rate capability is not satisfactory yet. In other words, the high capacity retention was sacrificed by high loading.

4) Does the electrode density include the current collector? What're the contents of Si, carbon and Ni in the composite?

5) In Figure 4b, the coulombic efficiencies of NG, SEAG and graphite are difficult to be distinguished.

6) Please test the long-term cycling stability at 1.75 mA cm⁻² for at least 500 cycles and that at 3.5 mA cm⁻² for at least 1000 cycles. These results would strongly support the good cycling stability of SEAG.

7) What's the loading mass of SEAG in electrode in terms of unit (mg/cm²)? If it is 10.5 mg cm⁻² as listed in table S1, the areal capacity at 1.75 mA cm⁻² is 525 mAh g⁻¹ x 10.5 mg cm⁻² = 5.5 mAh cm⁻² (Figure 4b), WHICH is much higher than 3.5 mAh cm⁻² shown in the text. So, this contradiction makes me confused.

8) How to understand that SAEG has the higher capacity retention than hard carbon? (Figure 4c) Although a-Si nanolayer might have more reactive sites than carbon, its would not guarantee that this capacity could be effectively kept at a high rate. The high capacity is more likely to be related to reaction kinetics. So, which one has the fast Li⁺ diffusion, a-Si or hard carbon? Please discuss this in detail.

9) If potentiostatic stages are responsible for Li plating, why not omit it from the charging? In other words, why do we need potentiostatic stages after galvanostatic stage?

10) Why graphite degrade so fast in full cell? If that is the result, LIBs won't be success now. Please check the CC-CV test of graphite and discuss it in detail. By the way, 5.1 mA cm⁻² is not a high current density for graphite in this case, in view of high loading (~10.5 mg cm⁻²).

Reviewer #3 (Remarks to the Author):

The manuscript is not of publication quality. Very basic and essential information and analyses are omitted. For instance, the surface area, density and basic composition of the EAG and SEAG materials are not reported. This does not allow a quantitative description of the electrochemistry of these materials in terms of their composition, which is not described in the manuscript even at the most basic level. Very important is the amount of carbon added to the graphite during the deposition

process. The electrodes are compared to a graphite coating containing a very small amount of carbon black. This electrode has a large impedance and would benefit from more carbon black. Therefore EAG should be compared to the effect of simply adding carbon black to the graphite electrodes. Further detailed remarks are below.

Experimental details are insufficient. Where applicable the purity, surface area, particle size, molecular weight etc. of all reagents needs to be disclosed. Some sources of chemicals are also missing (e.g. CMC, SBR, LCO, Li metal). The source/purity of reactive gases is missing. The type of x-ray radiation used is also not disclosed.

P12L303

Was anhydrous NiOH actually used?

The final electrode porosity after calendaring needs to be stated.

Grammar: the word "performances" needs to be replaced with "performance" in all cases.

P5L136

"The phase transition of the remaining Ni nanoparticles in EAG into Ni silicide"

This is a chemical reaction, not a phase transition.

The amount of Si, Ni and C deposited needs to be quantitatively determined. A good thing to do would be to weigh the graphite before and after each deposition step. It is a major weakness in this study that while many complex analyses have been conducted, basic experiments like simply measuring weight change have not.

Figure 4

The graphite capacity shown is very low compared to theoretical capacity of graphite. In addition the graphite plateaus are barely visible. Obviously the high loading, high densification and low carbon black content of this electrode have resulted in high electrode impedance and reduced capacity. Many readers will not understand this. The authors need to explain how the electrode formulation, loading and density can affect the capacity.

The differential capacity needs to be shown so that the Si and graphite contributions can be easily seen. Furthermore a complete analysis of the differential capacity curve needs to be discussed, with contributions from graphite, deposited carbon and silicon identified. This is another example of a basic and essential analysis that is missing.

The coulombic efficiencies should be plotted on a reasonable scale (e.g. from 98-100%). CE values less than this are not of interest.

P6L158

"The BET specific surface area of SEAG was estimated to be $2.49 \text{ m}^2 \text{ g}^{-1}$ "

What was the original surface area of the graphite? This needs to be mentioned. The densities of the graphite, EAG and SEAG need to also be mentioned.

P7L196

"Even under these challenging electrode conditions, SEAG achieved an outstanding CE (93.8%), with a high specific capacity (525 mAh g^{-1}), in the 1st cycle (Fig. 4a). To the extent of our knowledge, this is one of the highest value for the initial CE among previously reported graphite-Si composites."

There needs to be a complete analysis of how much Si is in these electrodes and if the Si content agrees with the observed capacity. For instance, if the capacity of the graphite is 323 mAh/g and the capacity of Si is 3579 mAh/g, then to raise the capacity to 500 mAh/g will require roughly 2.5 atomic % Si. Integration of the Si contribution to the differential capacity would be a more accurate method of determining the contribution of Si to the electrode capacity. Note that this is a very small amount of Si compared to previous studies. This should be taken into consideration when comparing CE values.

P8L208

"As higher current was applied, the lithiation capacity of graphite suffered from severe decay, and it exhibited only about 2% of the initial capacity (8 mAh g⁻¹) at a high current density of 10.5 mA cm⁻². On the other hand, the improved lithiation behavior was attained for EAG, and it provided higher lithiation capacity at current density of 7 mA cm⁻², over twice that of graphite (Supplementary Fig. 5)."

This is not surprising. Essentially, EAG is the same as graphite with carbon black added to the surface. As mentioned above, the graphite electrodes made in this study likely have high impedance because of the low amount of carbon black used. The amount of carbon added to the graphite by the EAG process needs to be reported. In order to claim that EAG has some advantage, it is essential to show that EAG achieves better performance than can be obtained by simply mixing more carbon black in the graphite coating.

SEAG should also be compared to conventional commercial graphite electrode performance (e.g. a Samsung ICR18650-32A high energy cell also can achieve 80 % charge capacity at 2C rate.). If SEAG represents a large improvement over conventional graphite, why can a commercially available high loading cell do the same thing?

Reviewers' comments:

Reviewer #1 (Remarks to the Author):

The authors have done a good job of responding to all of my original comments.

In some places, the English language should be improved.

Reviewer #2 (Remarks to the Author):

The revised manuscript and response letter address some of my concerns. But there are still a few questions for this paper.

1) Although the high mass loading and large areal capacity of electrode materials are very challenging for electrochemical tests, we still expect this work shows good cycling stability and rate capability. Otherwise, it would not be qualified for Nature Commun. For example, in the half cell, the poor cycling performance was attributed to Li anode that easily deteriorated upon cycling. This issue could be addressed by using a stable counter electrode, such as a commercial cathode material. But even in this case, the full battery still showed the inferior cycling stability. Why? For the rate performances, the unsatisfactory capacity retention was attributed to high areal current density (3C, $\sim 10.5 \text{ mA cm}^{-2}$). So, is it worth to develop such an electrode with a high areal capacity at the expense of rate capability and cycling stability?

2) As shown in Figure 4c and Figure R3, the delithiation of Si-Li was almost retained at the high current densities but that of graphite almost disappeared, how to understand this? In other words, all these results together with the poor rate capability both in half cells and in full cells were related to the high mass loading, high densification of graphite? Is it really necessary to achieve such a high capacity for anode materials, because this indicates an even higher mass loading of active materials on the side of cathode.

3) Referee1 question 6. The rough estimation on the capacity of SEAG composite does not take Ni ($\sim 2.8 \text{ wt\%}$) into accounts, no matter it is active within the tested potential window or not.

4) Figure R3 seems to be very different from Figure R14. Why?

Reviewer #3 (Remarks to the Author):

The authors have done a good job in their rebuttal. I have a few comments:

Please confirm if the NaOH used was anhydrous or actually NaOH·H₂O.

The authors should add an analysis comparing the electrochemical results found in Supplementary Figure 7 with the ICP-OES results shown in Table 1. I have found by my own calculation that these electrochemical results and ICP-OES results agree very well. This should be mentioned in the paper to strengthen both of these results.

Irrespective of how the EAG process works, it increases the total electrode surface area. Additions of carbon black do the same thing. Therefore the performance of EAG should be compared to simply

adding carbon black to graphite electrodes to increase the total electrode surface area to the same level as the EAG process.

REVIEWERS' COMMENTS:

Reviewer #2 (Remarks to the Author):

These authors have done a good job. There are no questions any more.

Reviewer #3 (Remarks to the Author):

The authors have done a good job in addressing all my comments. I encourage them to mention the newly obtained results showing that the performance of EAG is superior to adding equivalent amounts of carbon black (by surface area) to the manuscript or additional information section. I believe readers will be interested in this result.

Point by point response to the comments

Reviewers' comments:

Reviewer #1 (Remarks to the Author):

This paper reports a novel method for obtaining increased rate capability in graphite anodes. The approach is interesting, and the results shown here demonstrate that it is viable. I have several relatively minor criticisms:

Authors' response:

Thanks for your careful and insightful review of our manuscript. You bring up several interesting points that we hope we can adequately address here.

(1) The rationale for studying these materials is not particularly well written. Here, the authors should probably employ the services of a native English speaker.

Authors' response:

We appreciate the reviewer for carefully reading our manuscript and giving great advice. We have revised our manuscript and it was carefully proofread by professional.

(2) In their introduction, they do not say enough about other attempts to obtain faster kinetics by exposing graphite edges. A couple of references are mentioned (e.g., the engineered porosity in ref 12), but they should really discuss the idea of creating exposed edges here in more detail, since this is central to their work. Relevant prior studies include: Persson et al, J Phys Chem (2010), 1, 1176 and Mukhopadhyay et al, Adv. Functional Mater (2013), 23, 2397 (there is other pertinent work as well).

Authors' response:

Thank you for your insightful suggestion. We agree with that creating exposed edges is one of main concepts in our work and more detailed discussion about that should be provided.

We have carefully revised our manuscript to properly highlight the creating edge sites and added related citations.

(3) In several places, they emphasize the need for commercially viable approaches, yet the CVD approach seems potentially costly.

Authors' response:

In terms of cost estimation for CVD approach, we developed the customized rotatable CVD furnace in pilot level for 1 kg per batch (patent application number: KR101637980) (Figure R1). Moreover, both catalytic hydrogenation and a-Si/graphitic carbon deposition processes, which are based on the simple heat treatment

in specific atmosphere, can be conducted in one furnace sequentially. As a result, the simplification of this process can decrease the necessary time for synthesis, leading to a reduction in price. Therefore, we believe this process can be potentially applicable for the industry. We hope we have adequately addressed your concerns about our work.

[Figure R1 was redacted here]

(4) The authors do not really discuss SEI effects. Their surface treatment almost certainly alters the SEI, and this could be relevant. For example, the FEC added to the electrolyte is known to improve performance with Si containing electrodes, and this could be important. Have the authors looked at the behavior of their materials with any other electrolytes, to see if the improvements dependent on the electrolyte ?

Authors' response:

Thank you for your constructive suggestion. We totally agree with that the addition of additives can alter the SEI and improve electrochemical performance of our SEAG composite. As is well known, a continuous SEI formation derived from cracking during repeated cycles is considered as one of major problems for Si containing electrodes¹. To address this problem, the development of a stable SEI layer on Si surface have been achieved by introduction of various kind of electrolyte additives. In this regard, the usage of these additives is essential for Si containing electrodes.

To investigate the effect of various electrolyte additives on electrochemical behavior of SEAG, we have added both dQ/dV plot at the first cycle and the cycling performance of the SEAG composite along each additive to Supplementary Information (Figure S10) and in Figure R2 shown below. In figure R2 (a), it is clearly shown that the presence of additives suppresses the reduction of electrolyte such as EC^{2,3}. In addition, the SEAG composite exhibited good cycling performance even with relatively low content of FEC and VC additive (3 wt%). In case of baseline electrolyte which have no FEC and VC additive, the cycling performance

was gradually stabilized even though it was degraded early in the cycle.

Figure R2. Electrochemical properties of SEAG in various additive composition. (a) Differential capacity (dQ/dV) plot during lithiation process at the first cycle. (b) Cycling performance at current density of 1.75 mAh cm⁻². (c) Coulombic efficiency for cycles.

(5) The a-Si layer could slow down diffusion. Since this layer is thin, it does not appear to be a limiting factor (based on their results). However, this should be discussed more clearly.

Authors' response:

We agree with this reviewer that Si anode typically has lower Li ion diffusion coefficient than that of graphite and this fact could slow down diffusion in Si-graphite composite. However, the a-Si layer on our SEAG composite have homogeneously coated with 18 nm thick. Such thin Si nano-layer facilitates faster diffusion of Li ion by reducing the diffusion length. Therefore, this size effect of Si layer is advantageous to Li ion

diffusion. As proof, we have added differential capacity plot of SEAG composite and pristine graphite at various charge current density to Supplementary Information (Figure S12) and Figure R3. Since the lithiation potential of graphite and silicon is quite close, we analyzed the differential capacity for delithiation process to show more clear difference between samples. According to the figure R3(a) and (c), the oxidation region of graphite was reduced significantly as higher current applied. Whereas SEAG maintained relatively large oxidation area even at 10.5 mAh cm⁻². In particular, above 0.4 V, SEAG exhibited the additional oxidation area which indicates the delithiation reaction of Li-Si and such area was well retained even under high current density (figure R3(b) and (d)). It means that even if a high current was applied, the reactivity of Si does not change much in comparison with that of graphite. In this respect, it is clear that the a-Si layer on our SEAG composite is not a limiting factor in Li ion diffusion.

Figure R3. Differential capacity (dQ/dV) plot for delithiation process at various charge current density. All tests were measured by only the galvanostatic method. dQ/dV plot of (a), (b) SEAG and (c), (d) pristine graphite. Discharge current density was fixed at 1.75 mAh cm⁻².

(6) An estimate of the amount of Si added to the electrodes would be useful. A very thin Si layer on a dense particle of this size would presumably not provide this much increased capacity ? The porosity of the graphite is presumably important here ?

Authors' response:

We thank the reviewer for this constructive suggestion. In terms of the amount of Si content in our SEAG composite, above 6 wt% of Si should be included to achieve the specific capacity of 520 mAh g⁻¹.

$$\left({}^a 323 \text{ mAh g}^{-1} \times \frac{94}{100} \right) + \left({}^b 3600 \text{ mAh g}^{-1} \times \frac{6}{100} \right) \cong 520 \text{ mAh g}^{-1}$$

- a. Specific capacity of edge-activated graphite (EAG)
- b. Theoretical specific capacity of silicon

To prove this, we have added the element composition of the SEAG composite through inductively coupled plasma optical emission spectrometry (ICP-OES) to Supplementary Information (Table S1) and in Figure R4 shown below. According to this quantitative analysis, the amount of Si content is identified as 6.3 wt% in the SEAG composite.

In addition, we have provided the specific surface area of materials including EAG to Figure R5. Such specific surface area of EAG can sufficiently accommodate 6.3 wt% of Si layer with thickness of 18 nm as following calculation.

* Assume that the whole surface of EAG was covered by a-Si nanolayer with uniform thickness of 18 nm.

· *Total volume occupied by Si in 1 g of SEAG*

$$3.08 \text{ m}^2 \cdot \text{g}^{-1} \times 1 \text{ g} \times 18 \cdot 10^{-9} \text{ m} = 5.54 \cdot 10^{-8} \text{ m}^3$$

· *Maximum content of Si accommodated on 1 g of EAG*

$$5.54 \cdot 10^{-8} \text{ m}^3 \times 2.329 \text{ g} \cdot \text{cm}^{-3} \cong 0.129 \text{ g}$$

Specific surface area of EAG = 3.08 m² g⁻¹
 Thickness of deposited a-Si layer = 18 nm
 Volumetric mass density of Si = 2.329 g cm⁻³

As a result, the EAG can hold up to 11.4 wt% of 18 nm thick Si layer on its surface.

Sample	Element	Result	
SEAG	Si	6.3 wt%	ICP-OES
	Ni	2.8 wt%	Varian
	C	-	700-ES

Figure R4. ICP-OES measurement of SEAG composite.

Figure R5. Specific surface area of each sample.

(7) Given the large expansion and contraction of the Si during cycling, it's a little surprising that the material still functions reasonably well after 50 cycles. Did the authors try to do any post-cycling TEM (this is admittedly challenging) ?

Authors' response:

As described in our manuscript, SEAG has very thin a-Si layer with 18 nm thick and such layer is coated on the activated hole which provides void spaces to accommodate the volume expansion of Si for cycling. These features are believed to contribute to the excellent electrochemical performance of SEAG. As proof, we conducted the post-cycling TEM analysis for SEAG composite after 50 cycles as this reviewer suggested (Figure R6). As shown in Figure R6, the Si nanolayer is well adhered to the graphite without any delamination or crack of the layer and the deposited Si is also retained in a layer morphology very well without any aggregation of each other even after 50 cycles. These structural stabilities of SEAG during cycles result in the excellent cycling performance.

We have revised our manuscript and added this result to Supplementary Information (Figure S10)

Figure R6. TEM analysis of SEAG after 50 cycles. (a) HAADF-STEM image of silicon nanolayer on the activated hole. (b) HR-TEM image of the silicon nanolayer. The epoxy deposition on the sample was conducted to form a protective layer for TEM sampling.

Reviewer #2 (Remarks to the Author):

This work reported a high-performance anode material based on a-Si nanolayer deposited on edge-site activated graphite (SEAG). This anode material, SEAG, showed an excellent initial coulombic efficiency (~94 %), a large areal capacity (~ 3.5 mAh cm⁻²) and a high electrode density (~ 1.6 g cm⁻³). All these results are highly desirable, so very attractive for the industrial application. But some data could not support the results well.

Authors' response:

Thank you for your careful and insightful review of our manuscript. You bring up several important points that we hope we can adequately address here.

1) How about the output of one batch (g or Kg) and homogeneity of the product? CVD is an effective technique to deposit a layer on the surface of another material, but the homogeneity of as-obtain product is a great challenge. Because the gas penetration into the powders is limited, the particles nearly below or around the surface might be well coated. But those at the bottom have less chance to be coated. So, is there any solution to address this issue?

Authors' response:

We agree with this reviewer that achieving the homogeneity of as-obtained product is a great challenge for conventional CVD technique, especially in large batches. To address such homogeneity problem of traditional CVD process, we developed the customized rotatable CVD furnace in pilot level for 1 kg per batch (patent application number: KR101637980) (Figure R1) as discussed at the comment (3) of reviewer#1. To maximize the homogeneity of deposition layers, we customized the inner design of the furnace and adopted rotating system. As a result, we could obtain the product with uniform quality even in a kilogram batch. We hope we have adequately addressed your concerns about our work.

2) Since a-Si is directly exposed to environment, it would be heavily oxidized even at room temperature. Then, the resultant SiO_x would greatly reduce the initial coulombic efficiency, which seems to be opposite to the high value reported here (~94 %). So, please check the XPS spectra and discuss this in detail.

Authors' response:

The formation of native oxide layer on the Si surface is thermodynamically inevitable and such oxide layer would cause the irreversible phase such as Li₂O and Li₄SiO₄ during lithiation process. To verify the presence of native oxide layer in SEAG composite, we conducted X-ray photoelectron spectroscopy (XPS) as this reviewer suggested (Figure R7). This XPS result indicates that the native oxide layer exists on the surface and the thickness of this layer would be very thin. On the basis of the effective electron escape depth for Si

2p in elemental Si ($\sim 15 \text{ \AA}$)⁴, such native oxide layer is considered as thinner than 15 \AA otherwise the Si 2p peak cannot be detected. Therefore, this thin oxide layer do not have a significant impact on the initial coulombic efficiency. In the recent study, Gao et al. analyzed electrochemical behavior of silicon nanoparticles produced by a thermal CVD process⁵. Even though such Si nanoparticles also had native oxide layer due to the exposure to air, it achieved high initial coulombic efficiency of above 91%.

We have added this result to Supplementary Information (Figure S5) and revised our manuscript.

Figure R7. X-ray photoelectron spectra of the SEAG composite.

3) The fast charging of SEAG is not so good as the authors acclaimed in the manuscript. In view of the high areal capacity ($\sim 3.5 \text{ mAh cm}^{-2}$), the rate capability up to 10.5 mA cm^{-2} ($\sim 3 \text{ C}$) is not enough. Furthermore, the capacity retention dropped to 20 % of the initial capacity. These results indicate that the rate capability is not satisfactory yet. In other words, the high capacity retention was sacrificed by high loading.

Authors' response:

In a practical battery, the rate capability under the current density of 10.5 mA cm^{-2} is very challenging considering its hard electrode conditions including high areal capacity although it might seem insufficient when converted to C-rate. To the best of our knowledge, few works, if any, have reported the rate capability of anode materials under these challenging electrode conditions. As described in our manuscript, we also employed only the galvanostatic method during charging step in the rate test to examine the lithiation behavior which is highly affected by the over-potential. Especially in charging stage, the electrochemical behavior of graphite based anodes is significantly sensitive to the over-potential since their lithiation potential ($\sim 0.08 \text{ V}$ versus Li/Li^+) and cut-off voltage (0.005 V versus Li/Li^+ in this work) are very close. For this reason, the charging process would be ended before achieving fully lithiation without potentiostatic step. In this regard, it is not such a weird result that the low capacity is exhibited in the only galvanostatic measurement. For the better understanding, we have added the related caption to Figure 4(c).

4) Does the electrode density include the current collector? What're the contents of Si, carbon and Ni in the composite?

Authors' response:

The electrode density described in our manuscript does not include the current collector. It considers only the electrode materials including active material, binders and conductive agent.

To analyze the amount of such elements precisely, we conducted various kind of analyses including inductively coupled plasma optical emission spectrometry (ICP-OES) (Figure R4), thermogravimetric analyze (TGA) (Figure R8), and elemental analyzer (EA) (Figure R9). In the result of ICP-OES, we confirmed that the amount of Si and Ni were about 6.3 wt% and 2.8 wt%, respectively. Moreover, similar result was obtained with TGA analysis. However, carbon element could not be detected by this ICP analysis and it is also difficult to confirm the trace of carbon deposited on graphite using TGA. For these reasons, we tried to confirm the amount of deposited carbon using EA which generally used to quantitatively analyze organic elements. However, the carbon contents of EAG and graphitic carbon-deposited EAG were almost the same. This is because the amount of deposited carbon is smaller than the error range of our EA tool (1%). On the basis of these results, we can approximately specify that the amount of deposited carbon is less than 1%.

Figure R8. Thermogravimetric analysis of each sample.

Sample	Element	Result	
EAG	Carbon	96.3 wt%	EA Thermo Flash 2000
Carbon deposited EAG	Carbon	96.5 wt%	

Figure R9. Element analysis of EAG and carbon deposited EAG.

5) In Figure 4b, the coulombic efficiencies of NG, SEAG and graphite are difficult to be distinguished.

Authors' response:

We have added the magnified plot of coulombic efficiencies to Supplementary Information (Figure S9) and Figure R10.

Figure R10. Plot of the Coulombic efficiency of each sample for cycles.

6) Please test the long-term cycling stability at 1.75 mA cm⁻² for at least 500 cycles and that at 3.5 mA cm⁻² for at least 1000 cycles. These results would strongly support the good cycling stability of SEAG.

Authors' response:

We have added the cycling performance of the SEAG composite for 100 cycles at 1.75 mA cm⁻² and 140 cycles at 3.5 mA cm⁻² to Supplementary Information (Figure S8) and in Figure R11.

We agree with this reviewer that it would be good to show long-term cycling stability for 500 cycles or longer. However, in case of our highly loaded electrode, it is nearly impossible to achieve such long-term cycles because of the degradation of Li metal counter electrode. When the highly loaded electrode of > 3.0 mAh cm⁻² performed more than 50 cycles, the cycling performance depends on the condition of Li metal counter electrode because it requires high current density. Under such a high current density, the counter electrode of lithium metal forms irregular solid electrolyte interphase (SEI) layers on the anode during the first several cycles and the almost smooth lithium surface becomes rougher cycle by cycle, with an increase in active area and a decrease in resistance⁶. This causes the formation of Li dendrites and increase in the Li SEI resistance, which result in the failure of the counterpart (Li metal)⁷. In order to show reliable and repeatable data, we had shown the cycling performance of 50 cycles in the main text.

Figure R11. Cycling performance of SEAG at various current densities. (a) Cycle life of SEAG at 1.75 mA cm⁻². (b) At 3.5 mA cm⁻². Charge and discharge current density are the same.

7) What's the loading mass of SEAG in electrode in terms of unit (mg/cm²)? If it is 10.5 mg cm⁻² as listed in table S1, the areal capacity at 1.75 mA cm⁻² is 525 mAh g⁻¹ x 10.5 mg cm⁻² = 5.5 mAh cm⁻² (Figure 4b), WHICH is much higher than 3.5 mAh cm⁻² shown in the text. So, this contradiction makes me confused.

Authors' response:

The overall electrode information was already described in Table S2. The values in Table S1 indicated the EAG's. The mass loading level of SEAG is 7.1 mg cm⁻².

8) How to understand that SAEG has the higher capacity retention than hard carbon? (Figure 4c) Although a-Si nanolayer might have more reactive sites than carbon, its would not guarantee that this capacity could be effectively kept at a high rate. The high capacity is more likely to be related to reaction kinetics. So, which one has the fast Li⁺ diffusion, a-Si or hard carbon? Please discuss this in detail.

Authors' response:

We carefully inform that artificial graphite have been used as the basic material, not hard carbon in this work. As we responded to the comment (5) of reviewer#1, we agree with that Si anode typically has lower Li ion diffusion coefficient than that of graphite. However, the a-Si layer on our SEAG composite was uniformly deposited with very thin thickness of 18 nm. Such thin Si nano-layer facilitates faster diffusion of Li ion by reducing the diffusion length. Therefore, this size effect of Si layer is quite advantageous to fast Li ion diffusion. As proof, we have added dQ/dV plot of SEAG and pristine graphite at various charge current density to Figure R3. As the higher charge current density was applied, the oxidation area of pristine graphite significantly decreased. Whereas the oxidation area of Si on SEAG was almost retained even at the charge current of 10.5 mAh cm⁻². As a result, it is clear that the a-Si layer on our SEAG composite is more favorable to fast Li ion diffusion than that of pristine graphite.

9) If potentiostatic stages are responsible for Li plating, why not omit it from the charging? In other words, why do we need potentiostatic stages after galvanostatic stage?

Authors' response:

As is well known, Li ion diffusion in both cathode and anode electrode is the rate-determining step in the charging process of lithium ion batteries. Especially at high charging current, this sluggish Li^+ diffusion unavoidably causes the concentration polarization (over-potential) which brings about the rapidly increase of cell voltage to cut-off value and terminates the charging process before the cell is fully charged. For this reason, the potentiostatic stages (constant voltage mode) is essential to minimize the adverse effect of the concentration polarization and to fully charge the cell, particularly in fast charging.

In figure 4(d) of our manuscript, it was confirmed that pristine graphite and SEAG were charged only up to 29% and 68% at the charge current of 3.5 mA h cm^{-2} , respectively without the potentiostatic stage.

10) Why graphite degrade so fast in full cell? If that is the result, LIBs won't be success now. Please check the CC-CV test of graphite and discuss it in detail. By the way, 5.1 mA cm^{-2} is not a high current density for graphite in this case, in view of high loading ($\sim 10.5 \text{ mg cm}^{-2}$).

Authors' response:

As described in Table S2, the loading level of graphite is 12.1 mg cm^{-2} , which is a too challenging electrode condition to withstand high areal current density, along with high electrode density (1.6 g cm^{-3}) and low content of conductive agent (1%). In a recent research which is conducted by Argonne National Laboratory and BMW Group, it has proposed the guideline that graphite electrodes should avoid areal charging current density near or above 4 mA cm^{-2} to prevent severe side reaction and performance degradation⁸. According to this report, it can be confirmed that the cycle performance of cells adopted graphite anode is drastically deteriorated at the charging current density of 4.4 mA cm^{-2} . Likewise, in this work, the graphite cell has exhibited drastic capacity fading under such high charging current densities. For these reasons, we think that such rapid capacity degradation of graphite is natural, particularly in these challenging electrode conditions.

Reviewer #3 (Remarks to the Author):

The manuscript is not of publication quality. Very basic and essential information and analyses are omitted. For instance, the surface area, density and basic composition of the EAG and SEAG materials are not reported. This does not allow a quantitative description of the electrochemistry of these materials in terms of their composition, which is not described in the manuscript even at the most basic level. Very important is the amount of carbon added to the graphite during the deposition process. The electrodes are compared to a graphite coating containing a very small amount of carbon black. This electrode has a large impedance and would benefit from more carbon black. Therefore EAG should be compared to the effect of simply adding carbon black to the graphite electrodes. Further detailed remarks are below.

Authors' response:

Thank you for your careful and constructive review of our manuscript. You bring up several important points that we hope we can adequately address here.

1) Experimental details are insufficient. Where applicable the purity, surface area, particle size, molecular weight etc. of all reagents needs to be disclosed. Some sources of chemicals are also missing (e.g. CMC, SBR, LCO, Li metal). The source/purity of reactive gases is missing. The type of x-ray radiation used is also not disclosed.

Authors' response:

We would like to thank you for your constructive advice which improves the quality of our manuscript.

We have revised experimental details in our manuscript. The purity and source of all reagents used in this work are added, and the type of x-ray radiation is also specified (Cu-K α radiation).

2) P12L303

Was anhydrous NiOH actually used?

Authors' response:

We apologize for the confusion caused from omitting the purity of chemicals.

In this work, we did not use NiOH, but used NaOH of the purity > 98.0% and such experimental details have been added to our manuscript.

3) The final electrode porosity after calendaring needs to be stated.

Authors' response:

Thank you for your constructive point. The electrode porosity could be calculated as following formula.

$$\frac{(\text{True density of electrode}) - (\text{electrode density after calendaring})}{\text{True density of electrode}} \times 100$$

True density of carbon = 2.25 g cm⁻³

True density of silicon = 2.33 g cm⁻³

True density of nickel = 8.91 g cm⁻³

True density of CMC binder = 0.90 g cm⁻³

True density of SBR binder = 1.50 g cm⁻³

True density of electrode is

$$\left(2.25 \times \frac{90.9}{100} + 2.33 \times \frac{6.3}{100} + 8.91 \times \frac{2.8}{100}\right) \times \frac{96}{100} + 2.25 \times \frac{1}{100} + (0.90 + 1.50) \times \frac{1.5}{100} \cong 2.40 \text{ g cm}^{-3}.$$

Therefore, the final electrode porosity of SEAG is

$$\frac{2.40 - 1.60}{2.40} \times 100 \cong 33.3\%$$

We have revised our manuscript and informed the porosity with the electrode density.

4) Grammar: the word "performances" needs to be replaced with "performance" in all cases.

Authors' response:

We appreciate the reviewer for carefully reading our manuscript. We have revised our manuscript by following the reviewer's suggestions.

5) P5L136

"The phase transition of the remaining Ni nanoparticles in EAG into Ni silicide"

This is a chemical reaction, not a phase transition.

Authors' response:

Thank you for your great point. We have revised the phrase "phase transition" to "alloying reaction" in our whole manuscript

6) The amount of Si, Ni and C deposited needs to be quantitatively determined. A good thing to do would be to weigh the graphite before and after each deposition step. It is a major weakness in this study that while many complex analyses have been conducted, basic experiments like simply measuring weight change have not.

Authors' response:

We agree with your insightful suggestion that the amount of Si, Ni and deposited C should be quantitatively determined. As we answered to the comment (4) of reviewer#2, we conducted various kind of analyses including ICP-OES (Figure R4), TGA (Figure R8), weighing the samples before and after each step, and EA (Figure R9). In the result of ICP-OES and TGA, we confirmed that the amount of Si and Ni were about 6 wt% and 3 wt%, respectively. However, deposited carbon content could not be determined by ICP and TGA analysis. For these reasons, EA analysis of EAG and carbon-deposited EAG was attempted to determine the deposited carbon content. However, such carbon contents of EAG and carbon-deposited EAG were almost the same because the amount of deposited carbon is smaller than the error range of our EA tool (1%). On the basis of these results, we can approximately specify that the amount of deposited carbon is less than 1%.

Additionally, we have weighed the samples at each deposition step as this reviewer have suggested and the results was shown at Figure R12. It is noted that this method is unavoidable to occur errors during yield step. Unlike with the results obtained from ICP and TGA, the weight increase of Ni adsorbed graphite was measured a little higher and this difference seem to be derived from remaining chemicals which can be easily removed during heat treatment. In case of the carbon deposition process, the weight of carbon deposited EAG was rather decreased, since the amount of deposited carbon was much smaller than the error that can occur during yield step.

Process	Precursor	Product	Weight change
Ni adsorption	Graphite	Ni adsorbed graphite	+6.7%
Carbon deposition	EAG	Carbon deposited EAG	-2.2%
Silicon deposition	Carbon deposited EAG	SEAG	+4.9%

Figure R12. Weight change of the samples at each process.

7) Figure 4

The graphite capacity shown is very low compared to theoretical capacity of graphite. In addition the graphite plateaus are barely visible. Obviously the high loading, high densification and low carbon black content of this electrode have resulted in high electrode impedance and reduced capacity. Many readers will not understand this. The authors need to explain how the electrode formulation, loading and density can affect the capacity.

Authors' response:

We agree with this reviewer that the challenging electrode conditions can affect the electrochemical behavior of active material. In this work, we used artificial mesocarbon microbead (MCMB) graphite as pristine graphite which generally exhibited relatively low delithiation capacity ($\sim 320 \text{ mAh g}^{-1}$)⁹ than its theoretical capacity (372 mAh g^{-1}). For this reason, it is difficult to consider that the capacity is reduced due to such electrode conditions. However, the ambiguous plateaus of graphite can be derived from the electrode parameters including high loading, low amount of conductive carbon, and high densification. To verify the effect of such electrode conditions, we fabricated graphite electrode with various conditions (Figure R13). In case of low loading electrode (Figure R13(a)), the graphite plateaus which were barely visible in high loading electrode were clearly observed during the lithiation process.

Figure R13. Voltage profile of MCMB graphite at the first cycle with various electrode conditions. The electrode conditions varied (a) electrode loading level, (b) the content of conductive agent, and (c) electrode density after calendaring process. All other conditions are the same.

8) The differential capacity needs to be shown so that the Si and graphite contributions can be easily seen. Furthermore a complete analysis of the differential capacity curve needs to be discussed, with contributions from graphite, deposited carbon and silicon identified. This is another example of a basic and essential analysis that is missing.

Authors' response:

Thank you for your constructive point of our manuscript. We have added the differential capacity plot of SEAG and pristine graphite for delithiation process to Supplementary Information (Figure S7) and Figure R14. Furthermore, we conducted the integration of each plot, and calculated the area ratio and capacity contribution as different voltage range. As is well known, the capacity of graphite is mainly provided below 0.3 V and such capacity is around 310 mAh g⁻¹. Unlike the graphite, there is an additional oxidation region above 0.3 V in the SEAG. In this regard, it is clear that this additional capacity of SEAG above 0.3V is mainly contributed to Si and the capacity from Si is around 205 mAh g⁻¹. In terms of the capacity contribution of deposited carbon, it is negligible because the amount of deposited carbon is very small as we described in the comment (6) of this reviewer.

Sample	Area ratio		Capacity contribution	
	Below 0.3 V	Above 0.3 V	Below 0.3 V	Above 0.3 V
SEAG	60.9%	39.1%	319.7 mAh g ⁻¹	205.3 mAh g ⁻¹
Graphite	97.1%	2.9%	313.7 mAh g ⁻¹	9.3 mAh g ⁻¹

Figure R14. Differential capacity plot and capacity contribution of each sample.

9) The coulombic efficiencies should be plotted on a reasonable scale (e.g. from 98-100%). CE values less than this are not of interest.

Authors' response:

We have added the magnified plot of coulombic efficiencies to Supplementary Information (Figure S9) and Figure R10.

10) P6L158

"The BET specific surface area of SEAG was estimated to be 2.49 m² g⁻¹"

What was the original surface area of the graphite? This needs to be mentioned. The densities of the graphite, EAG and SEAG need to also be mentioned.

Authors' response:

We have revised Figure 2(c) and (d) to clarify the exact values of tap density and specific surface area. Also, we have added the density and specific surface area of EAG.

11) P7L196

"Even under these challenging electrode conditions, SEAG achieved an outstanding CE (93.8%), with a high specific capacity (525 mAh g⁻¹), in the 1st cycle (Fig. 4a). To the extent of our knowledge, this is one of the highest value for the initial CE among previously reported graphite–Si composites."

There needs to be a complete analysis of how much Si is in these electrodes and if the Si content agrees with the observed capacity. For instance, if the capacity of the graphite is 323 mAh/g and the capacity of Si is 3579 mAh/g, then to raise the capacity to 500 mAh/g will require roughly 2.5 atomic % Si. Integration of the Si contribution to the differential capacity would be a more accurate method of determining the contribution of Si to the electrode capacity. Note that this is a very small amount of Si compared to previous studies. This should be taken into consideration when comparing CE values.

Authors' response:

We appreciate the reviewer's insightful point. In Si–graphite composite system, the amount of Si is quite important factor because the electrochemical performance is very sensitive to such Si content. As we shown in Figure R4, Si content in our SEAG composite was identified as around 6.3 wt%. To carry out reasonable comparison of the initial CE value, we listed the previous reported results of Si–graphite composite which exhibited similar Si content and specific capacity with our SEAG composite (Figure R15).

Ref	Si content (wt%)	reversible capacity (mAh g ⁻¹)	I.C.E (%)
10	4.3	447	92.5
11	6.0	517	92.0
12	9.2	590	80.0
13	-	606	82.2
14	-	651	87.3
This work	6.3	525	93.8

Figure R15. Electrochemical behavior of Si–graphite composite anodes in previous works.

12) P8L208

"As higher current was applied, the lithiation capacity of graphite suffered from severe decay, and it exhibited only about 2% of the initial capacity (8 mAh g⁻¹) at a high current density of 10.5 mA cm⁻². On the other hand, the improved lithiation behavior was attained for EAG, and it provided higher lithiation capacity at current density of 7 mA cm⁻², over twice that of graphite (Supplementary Fig. 5)."

This is not surprising. Essentially, EAG is the same as graphite with carbon black added to the surface. As mentioned above, the graphite electrodes made in this study likely have high impedance because of the low amount of carbon black used. The amount of carbon added to the graphite by the EAG process needs

to be reported. In order to claim that EAG has some advantage, it is essential to show that EAG achieves better performance than can be obtained by simply mixing more carbon black in the graphite coating.

Authors' response:

We need to clarify the miscomprehension raised by the reviewer. In terms of performance comparison between EAG and pristine graphite, we applied exactly same amount of conductive agent (1 wt%) in all electrode fabrication processes. As we described in the manuscript and figure 1(a), the deposition process of graphitic carbon and a-Si were conducted in one furnace sequentially. **In this respect, EAG which is obtained right after the catalytic hydrogenation does not experience any carbon deposition process and it had no any additional conductive carbon compared with pristine graphite.** Therefore, it is reasonable that the better performance of EAG is derived from the enlarged edge planes.

Figure 1. (a) Schematic of the procedures for fabrication and characterization of SEAG.

13) SEAG should also be compared to conventional commercial graphite electrode performance (e.g. a Samsung ICR18650-32A high energy cell also can achieve 80 % charge capacity at 2C rate.). If SEAG represents a large improvement over conventional graphite, why can a commercially available high loading cell do the same thing?

Authors' response:

Thank you for your insightful suggestion of our manuscript. Unfortunately, it is not reasonable that comparison with between the commercial complete product such as Samsung ICR18650-32A and the single-layer pouch cell fabricated in lab scale. The properties of cathode and anode active material are generally regarded as one of main factors to determine the electrochemical behavior of the battery, but the cell manufacturing parameters also have considerable influence on the performance, in practical battery. According to the previous studies, the electrochemical performance of the battery would be significantly improved depending on cell design¹⁵, electrode fabrication sequences^{16, 17}, geometry of current collectors¹⁸, tab configuration^{19, 20}, electrode lamination, cell dimension^{21, 22}, and so on. For these reasons, it is difficult to compare our lab scale pouch cell with the state of the art commercial battery which has been completely optimized by major company.

In this work, in order to exclude such factors other than active material and demonstrate a practical viability of our SEAG composite, we compared this composite with MCMB graphite which has been successfully commercialized in battery industry, in the same condition.

References

1. Choi N-S, Yew KH, Lee KY, Sung M, Kim H, Kim S-S. Effect of fluoroethylene carbonate additive on interfacial properties of silicon thin-film electrode. *J Power Sources* **161**, 1254-1259 (2006).
2. Dalavi S, Guduru P, Lucht BL. Performance Enhancing Electrolyte Additives for Lithium Ion Batteries with Silicon Anodes. *J Electrochem Soc* **159**, A642-A646 (2012).
3. Xu K. Electrolytes and Interphases in Li-Ion Batteries and Beyond. *Chem Rev* **114**, 11503-11618 (2014).
4. Himpsel FJ, McFeely FR, Taleb-Ibrahimi A, Yarmoff JA, Hollinger G. Microscopic structure of the SiO₂/Si interface. *Physical Review B* **38**, 6084-6096 (1988).
5. Gao H, *et al.* Parasitic Reactions in Nanosized Silicon Anodes for Lithium-Ion Batteries. *Nano Lett* **17**, 1512-1519 (2017).
6. Lu D, *et al.* Failure Mechanism for Fast-Charged Lithium Metal Batteries with Liquid Electrolytes. *Adv Energy Mater* **5**, 1400993 (2015).
7. Nitta N, Yushin G. High-Capacity Anode Materials for Lithium-Ion Batteries: Choice of Elements and Structures for Active Particles. *Part Part Syst Char* **31**, 317-336 (2014).
8. Gallagher KG, *et al.* Optimizing Areal Capacities through Understanding the Limitations of Lithium-Ion Electrodes. *J Electrochem Soc* **163**, A138-A149 (2016).
9. Hossain S, Kim Y-K, Saleh Y, Loutfy R. Comparative studies of MCMB and C/C composite as anodes for lithium-ion battery systems. *J Power Sources* **114**, 264-276 (2003).
10. Li F-S, Wu Y-S, Chou J, Wu N-L. A dimensionally stable and fast-discharging graphite-silicon composite Li-ion battery anode enabled by electrostatically self-assembled multifunctional polymer-blend coating. *Chem Commun* **51**, 8429-8431 (2015).
11. Ko M, *et al.* Scalable synthesis of silicon-nanolayer-embedded graphite for high-energy lithium-ion batteries. *Nat Energy* **1**, 16113 (2016).

12. Khomenko VG, Barsukov VZ, Doninger JE, Barsukov IV. Lithium-ion batteries based on carbon–silicon–graphite composite anodes. *J Power Sources* **165**, 598-608 (2007).
13. Kim SY, Lee J, Kim B-H, Kim Y-J, Yang KS, Park M-S. Facile Synthesis of Carbon-Coated Silicon/Graphite Spherical Composites for High-Performance Lithium-Ion Batteries. *ACS Applied Materials & Interfaces* **8**, 12109-12117 (2016).
14. Uono H, Kim BC, Fuse T, Ue M, Yamaki JI. Optimized structure of silicon/carbon/graphite composites as an anode material for Li-ion batteries. *J Electrochem Soc* **153**, A1708-A1713 (2006).
15. Rieger B, Erhard SV, Kosch S, Venator M, Rheinfeld A, Jossen A. Multi-Dimensional Modeling of the Influence of Cell Design on Temperature, Displacement and Stress Inhomogeneity in Large-Format Lithium-Ion Cells. *J Electrochem Soc* **163**, A3099-A3110 (2016).
16. Dominko R, Gaberšček M, Drogenik J, Bele M, Jamnik J. Influence of carbon black distribution on performance of oxide cathodes for Li ion batteries. *Electrochim Acta* **48**, 3709-3716 (2003).
17. Kim KM, Jeon WS, Chung IJ, Chang SH. Effect of mixing sequences on the electrode characteristics of lithium-ion rechargeable batteries. *J Power Sources* **83**, 108-113 (1999).
18. Kosch S, Rheinfeld A, Erhard SV, Jossen A. An extended polarization model to study the influence of current collector geometry of large-format lithium-ion pouch cells. *J Power Sources* **342**, 666-676 (2017).
19. Li J, *et al.* 3D simulation on the internal distributed properties of lithium-ion battery with planar tabbed configuration. *J Power Sources* **293**, 993-1005 (2015).
20. Kim US, Shin CB, Kim C-S. Effect of electrode configuration on the thermal behavior of a lithium-polymer battery. *J Power Sources* **180**, 909-916 (2008).
21. Duan Y, Wu H, Huang L, Liu L, Zhang Y. Optimizing Current Terminals of 18 650 Lithium-Ion Power Batteries under High Discharge Current. *Energy Technology*, (2017).
22. Osswald PJ, *et al.* Current density distribution in cylindrical Li-Ion cells during impedance measurements. *J Power Sources* **314**, 93-101 (2016).

Point by point response to the comments

Reviewers' comments:

Reviewer #1 (Remarks to the Author):

The authors have done a good job of responding to all of my original comments.

In some places, the english language should be improved.

We would like to thank the reviewer's constructive comments for improving the quality of our work. Our revised manuscript has been proofread again by the native technical writer.

Reviewer #2 (Remarks to the Author):

The revised manuscript and response letter address some of my concerns. But there are still a few questions for this paper.

Authors' response:

We appreciate you spending the precious time and efforts on reviewing our manuscript.

1) Although the high mass loading and large areal capacity of electrode materials are very challenging for electrochemical tests, we still expect this work shows good cycling stability and rate capability. Otherwise, it would not be qualified for Nature Commun. For example, in the half cell, the poor cycling performance was attributed to Li anode that easily deteriorated upon cycling. This issue could be addressed by using a stable counter electrode, such as a commercial cathode material. But even in this case, the full battery still showed the inferior cycling stability. Why? For the rate performances, the unsatisfactory capacity retention was attributed to high areal current density (3C, $\sim 10.5 \text{ mA cm}^{-2}$). So, is it worth to develop such an electrode with a high areal capacity at the expense of rate capability and cycling stability?

Authors' response:

According to the recent studies which highlighted the true performance metrics of beyond-intercalation anodes, only the performance metrics containing all cell components can determine whether a true advance over conventional batteries has been achieved^{1, 2}. In terms of these performance metrics, high areal capacity and high electrode density are essential to realize high energy density in practical LIB configuration because they can raise energy density in pack-level while simultaneously lowering cost^{3, 4, 5}. In previous review, Obrovac et al. also suggested that the electrodes for energy cells should have high areal capacity ($> 2 \text{ mAh cm}^{-2}$), large amounts of active material ($> \sim 95\%$ by mass), low amounts of binder and carbon black, and low electrode porosity ($< 40\%$)⁶. However, most previous reports were simply studied under insufficient electrode conditions to demonstrate plausible electrochemical performance by using a lower areal capacity and higher electrode porosity than those of existing commercial electrodes.

With satisfying the practical electrode conditions, cycling the conventional battery over charging current density of 4 mA cm^{-2} is also a formidable challenge^{5,7,8}. According to the post-mortem study by Argonne National Laboratory, it clearly showed the trade-off relation between cycle stability and charging current density in conventional cells⁸. As higher current density applied, total resistance of cells increased by large concentration polarization which in turn causes thicker SEI formation. Such additional formation of SEI film consumes the limited Li^+ ions in full-cell and reduces the capacity. This performance degradation is primarily related to ionic conductivity and stability of electrolyte rather than characteristic of active material. In order to exclude such factors other than active material and demonstrate a competitive cycle stability of our SEAG composite, we have added the cycle performance of each sample at the low current density of 1.7 mA cm^{-2} to Figure R1 and Supplementary Information (Figure S17).

On the basis of these facts, we believe in the importance of our work in the development of fast rechargeable Li-ion batteries for the following reasons. As acknowledged by the reviewer, high areal capacity for the electrode with high densification is very challenging, especially at fast charging. Such high areal capacity is strictly limited by a critical charging current density which corresponds to a severe drop in cycle performance and in case of graphite electrode, its charging current density is restricted by about 4 mA cm^{-2} ⁵. For this reason, conventional graphite electrodes are still far from the fast charging application. In this regard, it is meaningful that our work overcomes the trade-off between cycle performance and charging current density and exhibits significantly improved performance without any signature of lithium plating compared to commercially available graphite, even under unprecedented current density over 10 mA cm^{-2} .

We would like to thank your insightful comment for improving the quality of our work, and we also hope

we have adequately addressed your concerns about our work.

Figure R1. Cycling performance of the full-cell with anodes of graphite and the SEAG composite under current density of 1.7 mA cm^{-2} . The cycling tests were performed in the voltage window between 2.7 and 4.35 V.

2) As shown in Figure 4c and Figure R3, the delithiation of Si-Li was almost retained at the high current densities but that of graphite almost disappeared, how to understand this? In the other words, all these results together with the poor rate capability both in half cells and in full cells were related to the high mass loading, high densification of graphite? Is it really necessary to achieve such a high capacity for anode materials, because this indicates an even higher mass loading of active materials on the side of cathode.

Authors' response:

The a-Si layer on our SEAG composite is homogeneously coated with 18 nm thick and such thin nano-layer can facilitate faster mass transfer by reducing the diffusion length. For this reason, very thin Si coating layer on SEAG is quite advantageous to Li ion diffusion. As a result, the delithiation area of Li-Si was well retained even at the high current density. (Fig. R3(b)) However, in case of graphite, its capacity contribution was rapidly decreased at high current density owing to the over-potential derived from its sluggish intercalation kinetics and low lithiation potential (0.08 V versus Li/Li⁺) close to cut-off voltage (0.005 V versus Li/Li⁺). Such over-potential and low lithiation potential terminate the lithiation reaction prematurely before fully lithiated in this rate test. (Fig. R3(c)) Whereas the edge-activated graphite in SEAG exhibited relatively large capacity contribution compared to that of pristine graphite due to its enhanced mass-transfer kinetics by activated edge-planes. (Fig. R3(a))

As we answered at above comment, high areal capacity (mass loading) and high electrode density are essentially required to meet the energy demand for future applications such as electric vehicles because they can easily increase energy density in pack-level with lowering manufacturing cost^{3,4,5}.

Figure R3 from previous response letter. Differential capacity (dQ/dV) plot for delithiation process at various charge current density. All tests were measured by only the galvanostatic method. dQ/dV plot of (a), (b) SEAG and (c), (d) pristine graphite. Discharge current density was fixed at 1.75 mAh cm^{-2} .

3) Referee1 question 6. The rough estimation on the capacity of SEAG composite does not take Ni (~2.8 wt%) into accounts, no matter it is active within the tested potential window or not.

Authors' response:

We apologized for the confusion from omitting the content of Ni in the estimation. On the basis of the ICP result, we have estimated the capacity of SEAG composite considering the content of Ni (2.8 wt%).

$$\left({}^a 323 \text{ mAh g}^{-1} \times \frac{90.9}{100} \right) + \left({}^b 3600 \text{ mAh g}^{-1} \times \frac{6.3}{100} \right) \cong 520.4 \text{ mAh g}^{-1}$$

- a. Specific capacity of edge-activated graphite (EAG)
- b. Theoretical specific capacity of silicon

According to this rough calculation, the capacity of SEAG in our experiment (525 mAh g^{-1}) agrees well with the expected capacity (520.4 mAh g^{-1}).

We have revised our manuscript to add this result to Supplementary Information (Figure S7).

4) Figure R3 seems to be very different from Figure R14. Why?

Authors' response:

The differential capacity plot in Figure R3 were measured by only the galvanostatic method, to analyze the lithiation behavior under high current densities. Whereas the result presented in Figure R14 was obtained by both galvanostatic and potentiostatic method, to investigate a capacity contribution of Si and graphite in SEAG composite. Because of such difference, the differential capacity profile in Figure R3 shows different shape compared with the result in Figure R14.

Reviewer #3 (Remarks to the Author):

The authors have done a good job in their rebuttal. I have a few comments:

Authors' response:

We appreciate the reviewer's insightful comments for improving the quality of our work.

1)Please confirm if the NaOH used was anhydrous or actually NaOH·H₂O.

Authors' response:

We apologize for the confusion caused from omitting the purity and grade of chemicals

In this work, we used anhydrous NaOH (> 98.0%, extra pure (EP) grade) in a bead form.

We have revised our manuscript to correct this information.

2)The authors should add an analysis comparing the electrochemical results found in Supplementary Figure 7 with the ICP-OES results shown in Table 1. I have found by my own calculation that these electrochemical results and ICP-OES results agree very well. This should be mentioned in the paper to strengthen both of these results.

Authors' response:

Thank you for your great suggestion. We have added related content in our revised manuscript.

3)Irrespective of how the EAG process works, it increases the total electrode surface area. Additions of carbon black do the same thing. Therefore the performance of EAG should be compared to simply adding carbon black to graphite electrodes to increase the total electrode surface area to the same level as the EAG process.

Authors' response:

We agree with this reviewer that the surface area of EAG electrode is higher than that of pristine electrode. To match the surface area of graphite electrode with the same level as the EAG electrode, we have applied additional amount of conductive agent (5.0 wt%) during the electrode fabrication of pristine graphite.

$$\text{Surface area of EAG electrode: } \left({}^a 3.08 \text{ m}^2 \text{ g}^{-1} \times \frac{96}{100} \right) + \left({}^b 62 \text{ m}^2 \text{ g}^{-1} \times \frac{1}{100} \right) \cong 3.58 \text{ m}^2 \text{ g}^{-1}$$

$$\text{Surface area of graphite electrode: } \left({}^c 0.51 \text{ m}^2 \text{ g}^{-1} \times \frac{92}{100} \right) + \left({}^b 62 \text{ m}^2 \text{ g}^{-1} \times \frac{5}{100} \right) \cong 3.57 \text{ m}^2 \text{ g}^{-1}$$

*The calculation considered the surface area of the active material and conductive agent only.

- a. Specific surface area of EAG
- b. Specific surface area of conductive agent
- c. Specific surface area of pristine graphite

We have added the electrochemical performance of the graphite electrode with additional carbon black in Figure R2. In the Fig. R2(a), the electrode containing more carbon black showed lower initial Coulombic efficiency (92.6%) than that of pristine electrode (1 wt% of carbon black). The decreased ICE of the electrode is derived from the excessive amount of carbon black which causes the side reaction with electrolyte⁹. In addition, such excessive amount of conductive agent brings about reduced energy density of the electrode by decreased active material ratio in whole electrode composition. According to the rate capability (Fig. R2(b) and (c)), even though additional amount of carbon black was applied, it still exhibited a lower galvanostatic capacity at various charging current density compared to that of EAG electrode which contains 1 wt% of carbon black. This result indicates that the rate-limiting factor of graphite for fast lithiation is mass-transfer kinetics rather than charge-transfer reaction^{10, 11, 12}. For this reason, it is reasonable that the EAG which has the enhanced mass-transfer kinetics by activated edge-planes showed the improved rate capability in comparison with the graphite containing excessive amount of carbon black. We hope we have adequately addressed your concerns about our work.

Figure R2. Voltage profile of EAG, pristine graphite, and graphite containing 5 wt% of carbon black at various current densities. (a) Voltage profile in the first cycle using both galvanostatic and potentiostatic method in charging step. Only galvanostatic charge/discharge profiles at (b) 3.5 mA cm⁻² and (c) 7.0 mA cm⁻².

References

1. Gogotsi Y, Simon P. True Performance Metrics in Electrochemical Energy Storage. *Science* **334**, 917-918 (2011).
2. Freunberger SA. True performance metrics in beyond-intercalation batteries. *Nat Energy* **2**, 17091 (2017).
3. De Volder MFL, Tawfick SH, Baughman RH, Hart AJ. Carbon Nanotubes: Present and Future Commercial Applications. *Science* **339**, 535-539 (2013).
4. Sun Y, Liu N, Cui Y. Promises and challenges of nanomaterials for lithium-based rechargeable batteries. *Nat Energy* **1**, 16071 (2016).
5. Gallagher KG, *et al.* Optimizing Areal Capacities through Understanding the Limitations of Lithium-Ion Electrodes. *J Electrochem Soc* **163**, A138-A149 (2016).
6. Obrovac MN, Chevrier VL. Alloy Negative Electrodes for Li-Ion Batteries. *Chem Rev* **114**, 11444-11502 (2014).
7. Buqa H, Goers D, Holzzapfel M, Spahr ME, Novák P. High Rate Capability of Graphite Negative Electrodes for Lithium-Ion Batteries. *J Electrochem Soc* **152**, A474-A481 (2005).
8. Somerville L, *et al.* The effect of charging rate on the graphite electrode of commercial lithium-ion cells: A post-mortem study. *J Power Sources* **335**, 189-196 (2016).
9. Marks T, Trussler S, Smith AJ, Xiong D, Dahn JR. A Guide to Li-Ion Coin-Cell Electrode Making for Academic Researchers. *J Electrochem Soc* **158**, A51-A57 (2011).
10. Takami N, Satoh A, Hara M, Ohsaki T. Structural and Kinetic Characterization of Lithium Intercalation into Carbon Anodes for Secondary Lithium Batteries. *J Electrochem Soc* **142**, 371-379 (1995).
11. Levi MD, Aurbach D. Diffusion Coefficients of Lithium Ions during Intercalation into Graphite Derived from the Simultaneous Measurements and Modeling of Electrochemical Impedance and Potentiostatic Intermittent Titration Characteristics of Thin Graphite Electrodes. *J Phys Chem B* **101**, 4641-4647 (1997).
12. Yu P, Popov BN, Ritter JA, White RE. Determination of the Lithium Ion Diffusion Coefficient in Graphite. *J Electrochem Soc* **146**, 8-14 (1999).

Point by point response to the comments

Reviewers' comments:

Reviewer #2 (Remarks to the Author):

These authors have done a good job. There are no questions any more.

Authors' response:

We would like to thank the reviewer's constructive comments for improving the quality of our work.

Reviewer #3 (Remarks to the Author):

The authors have done a good job in addressing all my comments. I encourage them to mention the newly obtained results showing that the performance of EAG is superior to adding equivalent amounts of carbon black (by surface area) to the manuscript or additional information section. I believe readers will be interested in this result.

Authors' response:

We appreciate the reviewer spending the precious time and efforts on reviewing our manuscript. We have added the result to Supplementary Figure 13.